# Optimizing 3D Laser Scanning Parameters for Early-Stage Defect Detectability in Subgrade Condition Monitoring

**DOI:** 10.3390/s25237174

**Published:** 2025-11-24

**Authors:** Mengmeng Liu, Gang Liu, Mingzhi Zhao, Xin Zhang, Kai Yang, Yang Chen

**Affiliations:** 1School of Architecture and Civil Engineering, Xihua University, Chengdu 610039, China; 0720090008@mail.xhu.edu.cn (M.L.);; 2Institute of Geotechnical Engineering, Xihua University, Chengdu 610039, China; 3China Metallurgical Group Limited, Chengdu 610063, China

**Keywords:** terrestrial three-dimensional laser scanning, highway embankment, point cloud, scanning parameters, surface reconstruction

## Abstract

Terrestrial three-dimensional laser scanning, which plays a crucial role in engineering surveying for assessing the surface smoothness of highway embankments by providing a level of precision and continuous three-dimensional information that conventional measurement methods cannot achieve, is examined in this study through a series of field experiments designed to determine how station location, including sampling interval, station distance, and scanning angle, influences point cloud density, spatial distribution, laser reflectivity, and surface reconstruction accuracy, and the results demonstrate that point cloud quantity decreases as sampling interval, station distance, and scanning angle increase, that the resolution of reconstructed surface undulations diminishes accordingly, that scanning angle has only a limited effect on reconstruction fidelity, that locating the instrument as close as feasible to the target area and adopting a sampling interval of 0.03 m achieves an effective balance between measurement accuracy and operational efficiency, and that optimizing parameter selection by analyzing elevation deviations at key points enhances both data quality and model precision, thereby confirming the suitability of the proposed approach for reliable highway embankment condition monitoring.

## 1. Introduction

In recent years, the durability of infrastructure worldwide, including roads and bridges, has declined rapidly due to natural disasters, creating an urgent need for efficient non-contact monitoring methods [1]. Traditional damage detection approaches, such as visual inspection and contact sensors, are time-consuming, pose health risks, and provide limited coverage, making timely maintenance and repair difficult [2]. Non-contact detection methods based on terrestrial three-dimensional laser scanning have therefore emerged as an effective alternative, offering detailed three-dimensional information without direct contact with the target [3,4].

In the field of civil engineering, terrestrial three-dimensional laser scanning technology has been increasingly applied to monitor deformation in highway and railway embankments, as it efficiently acquires detailed three-dimensional surface data and enables the detection of complex shapes and large-area targets [5,6,7]. This technology is particularly effective for identifying road surface defects such as cracks, repairs, potholes, and surface deformations. Digital elevation models constructed through interpolation algorithms allow the extraction of elevation profiles and longitudinal and transverse slope values to identify irregularities [8]. Fault values between concrete slabs can be calculated by measuring distances normal to reference planes to assess horizontal plane differences [9]. Terrestrial laser scanning can also measure longitudinal cracks in jointed concrete pavements, analyze slab curling and warping, and calculate average curvature [10]. Mobile laser scanning data can be processed to detect hazards such as potholes, heaves, and bumps, classify pavement segments based on elevation deviations, and assess rigid pavement slab defects, characterizing them according to defect area, crack width, and strength [11,12]. Table 1 summarizes the equipment, evaluation categories, and application scenarios used for assessing road structures with three-dimensional laser scanning devices.

Although previous studies have provided useful insights into road surface damage monitoring, significant challenges remain in achieving high-precision and high-efficiency scanning in complex environments [13,14,15] because the performance of terrestrial three-dimensional laser scanning is controlled both by object characteristics, including surface roughness, color, and shape, and by scanning conditions, including laser incident angle, laser range, sampling interval, and station height [16], and although earlier research primarily examined how factors such as color, roughness, incident angle, distance, and material type influence point cloud quality [17,18], showing, for example, that point cloud quantity decreases with distance when scanning sketch paper, that color strongly affects reflectivity with white surfaces producing the highest return, that laser incident angle exhibits a negative correlation with reflectivity, and that different materials such as standard reflectors and wood behave differently [19], and further explored the relationship among plane residuals, scanning distance, and incident angle to establish criteria for these parameters through numerical simulations and real data analysis based on plane fitting and the input precision of point spacing [20], most existing studies have relied on small vertical surfaces and single-factor analyses. Table 2 summarizes the equipment, evaluation parameters, and related details used for the assessment of these research subjects based on three-dimensional laser scanning devices.

This study investigates the use of terrestrial three-dimensional laser scanning for assessing the smoothness of highway embankment fill surfaces and focuses on optimizing scanning parameters for large horizontal surface measurements. The Leica MS60 total station three-dimensional laser scanner is used to survey the embankment, and the effects of sampling interval *Δ*, station distance *L*, and scanning angle *β* on scanning quality are evaluated by analyzing point cloud quantity, surface model resolution, and surface area. Delaunay triangulation algorithms are applied to reconstruct the embankment surface in three dimensions, and an optimized method for smoothness detection is developed. The study identifies an optimal combination of scanning parameters and extends previous research by applying the approach to large horizontal surfaces in real field settings, providing new insight into the practical application of terrestrial three-dimensional laser scanning in embankment monitoring.

## 2. Methodology

### 2.1. Three-Dimensional Laser Scanning Technology

#### 2.1.1. Point Cloud Coordinate Acquisition

Three-dimensional scanning technology operates on the principle of laser ranging, rapidly recording dense 3D coordinates across the target’s surface to construct a detailed 3D model [21]. The 3D scanner emits laser light, which diffusely reflects upon contacting the target object and is then received by the receiver. By recording the time interval between emission and reception, the distance between the target and the 3D laser scanner is calculated using the speed of light [22]. Using laser ranging as a foundation, the horizontal angle *α* and vertical angle *θ* of the laser beam are measured using the horizontal and vertical angle sensors within the instrument. These angles, along with the position information of the scanner, enable the measurement of three-dimensional coordinates on the target surface [23], as shown in Figure 1. The 3D coordinates of sampled points, known as the point cloud, are rapidly and densely acquired. In this manner, the three-dimensional coordinates of numerous sampling points, i.e., point clouds, can be quickly obtained [24]. The coordinates (*X*, *Y*, *Z*) of point *P* in 3D space can be calculated as(1)$$\left\{\begin{array}{c} X=S\cos\theta \cos\alpha \\ Y=S\cos\theta \sin\alpha \\ Z=S\sin\theta \end{array}\right.$$
where *α* is the horizontal scanning angle, *θ* is the vertical scanning angle, and *S* is the distance between the target and the scanner.

#### 2.1.2. Factors Affecting Scanning Accuracy

When a 3D laser scanner, including terrestrial laser scanning, is used for embankment surface detection, the quality of the point cloud is controlled by point cloud density, the inclination of the reflective surface, surface roughness, incident angle, and site conditions. Soil type, moisture content, lighting, and atmospheric conditions can also influence laser reflectivity and the measured response. These factors jointly determine the quality, density, and accuracy of the point cloud data, which affects the reliability of embankment surface smoothness and defect detection.

(1)Point cloud density

When a 3D laser scanner is used to assess the quality of a roadbed fill surface, point cloud data sparsity significantly impacts modeling accuracy [25]. Adequate point cloud density is essential for effective data acquisition of the fill surface and subsequent data thinning and compression [26]. The interval of the 3D laser scanning point cloud is a configurable parameter. The resolution of the 3D laser point cloud can be expressed by the point interval *dl* on the scanning plane, which can be expressed as:(2)$$dl=dz\cot\theta =\frac{sd\theta }{\sin\theta \cos\theta \cos\alpha }$$
where *r* is polar distance, *θ* is horizontal scanning angle, *s* is scanning distance, and *α* is vertical scanning angle. Additionally, the number of points per square meter on the vertical surface at a distance *s* from the scanner is(3)$$n=\frac{\cos^{3}\cos\theta \sin\theta }{s^{2}d\alpha d\theta }$$

The point cloud density decreases rapidly with increasing scanning distance, following an inverse square relationship. As the horizontal scanning angle increases, the number of points decreases and is proportional to the cube of the cosine of the angle. Similarly, as the vertical scanning angle increases, the point count decreases proportionally to the square of the cosine of the angle. Consequently, increasing scanning distance and horizontal angle reduces point cloud density. Excessively high point cloud density increases data acquisition and processing time without significantly improving accuracy.

(2)Reflective surface tilt

The laser scanning ranging system consists of two parts: a laser transmitter and a laser receiver. Since laser emission and reception share a common optical path, and the laser beam has a certain divergence angle, a laser foot spot is formed when it is scanned on the surface of an object [27]. There is a relationship between the size *d* of the laser foot spot, the laser emission aperture *D*, and the laser emission beam divergence angle *γ*, as follows(4)$$d=D+2S\tan\frac{\gamma }{2}$$

When the scanning target object is tilted, the normal line of the tangent plane on the scanning target object surface does not coincide with the direction of the laser beam, causing a deviation in the position of the laser foot point (Figure 2), as follows(5)$$ds=S_{1}-S=-\frac{S\gamma \tan\frac{\gamma }{2}}{2}$$

(3)Target surface roughness

The accuracy of 3D laser scanning is affected by the roughness of the target surface. When a laser beam is emitted, if the surface of the scanned target is complex, the echo signal will be a combination of some very small surface reflection waves [28]. For a single laser signal, the distance error caused by the measured surface is about half of the extreme value of the surface roughness.

(4)Incidence angle

When the laser is not projected perpendicularly to the incident surface, the laser and the incident surface will intersect as an ellipse, which is the laser incident angle deviation. In practice, the influence of the deviation can be offset by taking a weighted average of all reflected beams [29].

(5)Soil Type

Different soil types have distinct physical and chemical properties that influence the reflection of laser beams. Rough surfaces, such as gravel embankments or uncompacted coarse-grained soils, often cause diffuse reflection, scattering the laser signal, which can reduce point cloud density but capture surface details. Soil color also affects reflectivity, with dark-colored soils, such as black or wet soil, absorbing more laser energy and producing weaker reflected signals, which can decrease the number of valid point cloud points. Mineral composition and organic matter content further influence the interaction between the laser and the surface [30,31,32,33].

(6)Moisture Content

Moisture is a key environmental factor affecting the quality of LiDAR data. Water strongly absorbs laser energy, particularly at near-infrared wavelengths, and alters soil surface roughness and reflectivity. A thin water layer on the embankment surface can cause multiple reflections between the water and the underlying surface, producing multi-path effects that lead to noisy or inaccurate point cloud data [34,35,36].

(7)Lighting Conditions

Uneven lighting, including shadows or direct sunlight, can distort point cloud color information, affecting visual appearance and the identification of features based on color [37,38].

(8)Atmospheric Effects

Atmospheric conditions influence the propagation and reception of the laser beam. During transmission, the beam may be absorbed, scattered, or refracted by the atmosphere. Variations in atmospheric density, such as temperature or humidity gradients, can bend the laser beam. Strong winds may not affect laser propagation directly but can induce vibrations in the scanner or slight movement of the target, introducing measurement errors and reducing point cloud registration accuracy and the precision of the reconstructed model [39].

### 2.2. Experimental Procedures

#### 2.2.1. Testing Site

In order to analyze the influence of station distance and scanning angle parameters on scanning accuracy, a section of roadbed fill was selected as the testing ground. This section is located at a road intersection, measuring 100 m in length and 60 m in width. The site is sufficiently wide to meet various parameter settings. A rectangular area of 20 m × 10 m was selected as the scanning area, as shown in Figure 3. A rectangular area with a length of 20 m and a width of 10 m is taken as the scanning area. The field experiments were conducted under the same observational conditions, and the influence of external factors on the measurement results was minimal.

#### 2.2.2. Instruments

The Leica MS60 total station 3D laser scanner was employed for the test due to its dual functions of 3D scanning and total station capabilities, enhancing efficiency in scanning and measurement tasks. The Leica-MS60 uses a pulsed red laser with a wavelength of 658 nm, capable of 360° horizontal and 270° vertical rotation, providing a wide field of view. Its single-point ranging accuracy reaches up to 1.2 mm within 100 m, with angular accuracy up to 0.5″. The scanner achieves speeds of up to 1000 points per second, enabling high-speed scanning. Additionally, it features a point cloud visualization function for real-time data observation.

#### 2.2.3. Test Procedure

Firstly, the Leica MS60 was used to perform 3D scanning of the test area to obtain point cloud data. Then, the fill surface model was reconstructed. Finally, statistical analysis was performed on the number of points and the surface area of the reconstructed model to explore the effects of sampling interval, station distance, and scanning angle on the quality of fill surface scanning. The parameter settings are shown in Figure 4. The sampling interval (*Δ*), defined as the distance between the reference point and neighboring point clouds, was set to 0.01 m, 0.03 m, 0.10 m, and 0.30 m. The station distance (*L*), defined as the relative distance between the setup point and the midpoint of the long side of the scanning area, was set to 5 m, 15 m, and 30 m. The scanning angle (*β*), the angle between the setup point and the line to the midpoint of the long side of the scanning area, was set to 30°, 60°, and 90°. The test reference point was set at the center of the scanning area, and 18 scans with different parameters were performed. The scanning conditions are listed in Table 3.

During the test, it is necessary to establish an assumed coordinate system, which is formed by arbitrarily assuming the coordinates of a point and the azimuth of one side [40]. The test area was defined with the long side of the target area as the *X*-axis, positive to the east, and the short side as the *Y*-axis, positive to the south. Firstly, assume the coordinates of control point 01 and use the Leica-MS60 total station to measure both the relative distance between control points 01 and 02 and the elevation of control point 02. Then, connect points 01 and 02 to establish a known edge, measure the azimuth of this edge (assuming it is 200°), and calculate the coordinates of control point 02. Thus, the assumed coordinate system was established. Subsequently, the total station 3D laser scanner was positioned at the setup site for scanning, and the setup site coordinates were measured using the rear rendezvous principle. Following the scanning process, the total station function was employed to measure the coordinates of the four vertices of the test area, aiding in boundary determination during the post-processing of point cloud data.

### 2.3. Model Reconstruction Methods

The fill surface model is reconstructed using Delaunay triangulation, which connects pairs of points from the measured point set *P* with straight lines. This process forms a convex polygon as the outer boundary, creating an inhomogeneous triangular mesh known as the Delaunay triangular mesh [41], as shown in Figure 5. The Delaunay reconstruction algorithm initially performs triangulation, filtering triangles based on predetermined conditions to complete the surface reconstruction of the point cloud. This process begins with the creation of an initial triangle and iteratively adds and optimizes points across the entire point cloud [6,42,43].

#### 2.3.1. Delaunay Triangulation Principle

The first principle is to maximize the minimum angle. Among all triangles formed by a scattered set of points, Delaunay triangulation ensures that the minimum angle of triangles is maximized, making it the closest to a regular triangulation. This property ensures that the minimum interior angle does not increase when two neighboring triangles forming the diagonal of a convex quadrilateral are swapped, as shown in Figure 6a. The second principle is the empty circumscribed circle criterion, which states that for any triangle in the Delaunay triangulation of a point set *S*, its circumcircle does not contain any other points from the set *S*, as shown in Figure 6b. Another principle is uniqueness. For a point set S on a plane where no more than four points are cocircular, the Delaunay triangulation is unique. This means that regardless of the order in which the triangles are constructed from the points in the region, the resulting Delaunay triangulation remains consistent [44,45].

#### 2.3.2. Delaunay Triangulation Algorithm

KD-Tree’s full name is K-Dimensional-Tree, proposed by Bentley in 1975. KD-Tree is a data structure for managing k-dimensional spatial data, which is essentially a binary index tree with constraints. For point cloud data, it is a three-dimensional KD-Tree. It adopts the idea of divide-and-conquer, which can be regarded as dividing the k-dimensional space, with hyperplanes perpendicular to the coordinates in the specified dimensions. All child nodes correspond to child nodes in the space, and each non-leaf node represents a hyperplane, which is perpendicular to the coordinate axis of the currently divided dimension and divides the space into two parts in that dimension, with the left side of the hyperplane being the left subtree of the KD-Tree and the right side being the right subtree of the KD-Tree.

The division of KD-Tree is based on a number of factors, including the division of variance of each axis and the division of alternating cycles of each axis, the variance of the data points is usually chosen to split the node, the variance of the data points in k-space is calculated in each dimension, and the one with the largest variance is taken as the split-axis, and then the values are arranged in the split- axis, the values are then ranked and the middle value is used as the cut-off point.

The algorithm flow is shown in Figure 7. Firstly, the KD-tree of the point cloud data is constructed, and the k-neighborhood points of each point are searched. The average distance between each point and its neighboring points is then calculated and used as the local density parameter. Next, the 3D Delaunay triangulation of the point cloud data is performed. After obtaining the tetrahedral data structure, it is converted into a triangular structure. Overlapping triangles must be eliminated from the converted triangles. The radius of the circumcircle for all triangles, after eliminating overlaps, is calculated based on the marked index of the point cloud using the following formulas:(6)$$p=\frac{e_{1}+e_{2}+e_{3}}{2}$$(7)$$a=\sqrt{p(p-e_{1})(p-e_{2})(p-e_{3})}$$(8)$$r=\frac{e_{1}+e_{2}+e_{3}}{4a}$$
where *a* is the area of the triangle, and *r* is the radius of the circumcircle. Finally, a threshold is set based on the local range density of each triangle. If the circumcircle radius of a triangle exceeds this threshold, the triangle is eliminated. This process results in the reconstructed surface of the point cloud.

## 3. Results and Discussion

### 3.1. Scanning Data Results and Discussion

#### 3.1.1. Distribution Patterns of Point Clouds

The point cloud data was processed using the 3DReshaper (version 2018) software. Initially, the boundary of the test area was determined by importing the coordinates of the four vertices into 3DReshaper, forming four independent point clouds. These points were then connected with a polyline within the software. Subsequently, the point cloud was denoised by utilizing the cleanup function, which allowed for the selection of the test area and the removal of extraneous noise with a single click.

Figure 8 shows point clouds at different sampling intervals with a scanning angle of 90° and a station distance of 5 m. It shows that the point clouds are denser near the setup location and become sparser with increasing distance, showing an arc-shaped expansion and symmetrical distribution. The arc-shaped expansion occurs because the scanner uses an equirectangular scanning mode, which increases the interval between sampling points as the distance from the scanner grows. The symmetrical distribution results from the 90° scanning angle, with the setup station positioned on the centerline of the test area, equidistant from the left and right edges of the target area.

The distribution of point clouds at different station distances is shown in Figure 9, with a sampling interval of 0.03 m and a scanning angle of 90°. The point clouds are denser near the station and become sparser with increasing distance. As the station distance increases, the point cloud distribution becomes progressively sparser. The test area is divided along the centerline in the width direction. When the station distance is 5 m, the number of points near the setup site is approximately 3 times that of the point clouds farther away. However, when the station distance increases to 30 m, the number of point clouds near the setup site decreases to 1.2 times that of the point clouds farther away. This occurs because the test area has a width of 10 m, and increasing the station distance reduces the increase in scanning distance, thereby decreasing the variation in the sampling point interval during scanning.

As shown in Figure 10, with a sampling interval of 0.03 m and a station distance of 5 m, different scanning angles affect point cloud density. The point clouds are denser near the setup station and sparser farther away, but the uniformity of their distribution varies significantly. The test area is divided into two parts along the centerline in the length direction. At a 30° angle, the right part of the test area has significantly higher point cloud density than the left part, with 60,036 and 13,864 points, respectively. At a 60° angle, the density is more even, though still inconsistent, with 30,451 points in the right part and 17,087 points in the left part. At a 90° angle, the point cloud density distribution becomes more uniform, showing an axisymmetric distribution along the centerline, with 23,272 points in the left part and 23,871 points in the right part, differing by only 2.5%. The uneven point cloud distribution at scanning angles of 30° and 60° is mainly due to the inconsistent distance between the left and right parts of the test area and the setup site. The increased distance results in larger spacing between sampling points, reducing the number of point clouds in the left part of the test area.

#### 3.1.2. Statistical Analysis of Point Cloud Quantity

The number of points is the direct measurement data acquired by 3D laser scanning and serves as the basis for subsequent data analysis. Therefore, the point clouds obtained at different sites were statistically analyzed according to various influencing factors, as shown in Figure 11.

To investigate the effect of sampling interval on point cloud quantity, a statistical analysis was conducted, as shown in Figure 11a. The results demonstrate that sampling interval significantly influences point cloud quantity. When station distance and scanning angle are held constant, point cloud quantity decreases following a power function as the sampling interval increases. At a sampling interval of 0.01 m, the point cloud quantity is approximately 1000 times greater than that at 0.30 m, exhibiting a pronounced decreasing trend. The standard model for the power function is:(9)$$y=ax^{b}$$
where *a* and *b* are the fitting parameters. The data were fitted using a power function; the correlation coefficient is *R*^2^ = 0.98, and the fitting results indicate an excellent agreement with the observed data [46].

The statistical results indicate that the sampling interval, station distance, and scanning angle all influence the number of points. Among these factors, the sampling interval and station distance have a greater impact, while the scanning angle has a smaller effect in Figure 11b,c. Under the same conditions of sampling density and scanning angle, the number of points decreases logarithmically with increasing station distance. For instance, the number of points at a station distance of 5 m is about four times that at 30 m, demonstrating a substantial reduction trend, though less pronounced than that of the sampling interval. Additionally, when the sampling density and station distance are constant, the number of points decreases with increasing scanning angle, following a second-order polynomial function. However, this decrease is not significant, with the number of points at an angle of 30° being approximately 1.4 times that at 90°. The data were fitted using a second-order polynomial function, and the fitting model can be expressed as:(10)$$y=ax^{2}+bx+c$$
where *a*, *b* and *c* are the fitting parameters. The data were also fitted using a power function; the correlation coefficient is *R*^2^ = 0.99, and the fitting results show excellent agreement with the observed data.

#### 3.1.3. Discussion of Scanning Parameter Settings

Due to the limited vertical field of view of the 3D laser scanner, a scanning blind zone centered at the setup site with a radius *d* exists. If the minimum scanning angle *η* and working height ℎ of the 3D laser scanner are known, the radius *d* of the scanning blind zone can be calculated as follows(11)d=htanη
where *η* is the angle between the laser transmitter and the vertical direction, and ℎ is the height of the laser transmitter from the ground, as shown in Figure 3b. The vertical field of view angles of different 3D laser scanners vary, affecting the minimum scanning angle *η*. The Leica-MS60 total station 3D laser scanner used in this test can rotate 270° vertically, with a vertical angle range of 45° to 315°, and a minimum scanning angle *η* of 45°. The working height ℎ of the instrument in the test is 1.8 m.

Table 4 lists the number of points in the target area and the time required to complete the scanning work under various conditions, including station distances of 5 m, 15 m, and 30 m, and sampling intervals of 0.01 m, 0.03 m, 0.10 m, and 0.30 m. The data show that both the number of points and the scanning time decrease with increasing sampling interval and station distance. When the sampling interval exceeds 0.10 m, high-quality point cloud data cannot be obtained even at a station distance of 5 m. For shorter station distances, increasing the sampling interval reduces scanning time while maintaining accuracy. Conversely, for longer station distances, reducing the sampling interval improves scanning accuracy but increases scanning time and reduces efficiency. Therefore, the setup station should be positioned as close as possible to the target area while meeting the minimum scanning distance to balance scanning accuracy and efficiency.

To evaluate point cloud reliability, five checkpoints, including boundary points A, B, C, D, and central point E (Figure 3a), were selected within the scanned area. Elevation values extracted from the point cloud were compared with precise leveling measurements to assess accuracy. Elevation deviation data at these points were analyzed to evaluate three-dimensional point cloud accuracy. Comparison of deviation data with root mean square error (RMSE) values indicates a high level of agreement. When station distance and scanning angle are constant, elevation deviations are minimal at sampling intervals of 0.01 m and 0.03 m, as listed in Table 5. Considering the excessive scanning time at 0.01 m, a sampling interval of 0.03 m is recommended. Table 6 and Table 7 further analyze the effects of station distance and scanning angle. With a sampling interval of 0.03 m, elevation deviations indicate that the optimal station distance is 5 m and the optimal scanning angle is 90°. Based on the field trials, when the station distance is 5 m, the scanning angle is 90°, and the sampling interval is 0.03 m, the terrestrial laser scanner produces reliable point cloud data.

### 3.2. Model Reconstruction Analysis and Discussion

#### 3.2.1. Reconstruction of Fill Surface Model

Fill surface reconstruction involves projecting the point cloud onto a two-dimensional plane and constructing a triangular grid using the Delaunay triangulation algorithm. Elevation values from the point cloud are integrated to form a three-dimensional model, as shown in Figure 12. Under a scanning angle of 90° and a station distance of 5 m, the model accuracy in reflecting terrain fluctuations decreases significantly as the sampling interval increases. Consequently, the model resolution also diminishes. When the sampling intervals are 0.01 m and 0.03 m, the reconstructed model accurately describes the morphological characteristics of the fill surface, with the highest resolution observed at the 0.01 m interval. However, at sampling intervals of 0.10 m and 0.30 m, the error between the reconstructed model and the actual fill surface morphology increases significantly. Additionally, an evident missing area at the boundary of the target area becomes more pronounced with larger sampling intervals. This is due to the point cloud gaps at the test area boundary resulting from larger sampling intervals, which hinder the digital fill surface model from fully covering the target area. Figure 13 presents fill surface reconstruction models at different station distances, using a sampling interval of 0.03 m and a scanning angle of 90°. The resolution of the reconstructed surface decreases as station distance increases. The circular area highlights a concave surface, while the rectangular area shows rut marks left by vehicles. These features become increasingly blurred at greater station distances, reducing the model’s ability to represent actual terrain. Figure 14 shows fill surface reconstruction models at different scanning angles, using a sampling interval of 0.03 m and a station distance of 5 m. All models fully cover the test area and capture surface undulations with consistent resolution. This indicates that variations in scanning angle have minimal effect on the reconstruction of the fill surface.

The surface difference results of the reconstructed embankment models at different sampling intervals are shown in Figure 15. The comparison indicates that smaller sampling intervals yield higher reconstruction accuracy. In Figure 15a (sampling interval of 0.01 m), 92.5% of the deviations fall between −5 mm and −2.5 mm, showing that surface deformation remains small and point cloud deviations are highly concentrated. The models produced with sampling intervals of 0.01 m and 0.03 m exhibit similar accuracy. In Figure 15b, only 69.6% of the deviations fall within this range, and in Figure 15c, the proportion further decreases to 43.9%. These results demonstrate that larger sampling intervals cause greater surface deformation and reduce reconstruction accuracy. Finer sampling intervals (0.01 m and 0.03 m) allow more reliable detection of small elevation changes, whereas larger intervals (0.10 m and 0.30 m) improve efficiency but increase reconstruction errors. In summary, small sampling intervals enhance surface accuracy, while large intervals reduce precision and produce coarse models.

#### 3.2.2. Surface Area Analysis of Reconstructed Models

The reconstructed model serves as fundamental data in 3D laser scanning, where surface area is a crucial metric. Therefore, the surface area of the fill surface reconstructed model underwent statistical analysis. Due to the relatively flat terrain of the test area with minimal topographic variations, the surface area of the reconstructed model closely approximated the actual surface area. To emphasize the impact of various factors on the reconstructed fill surface model’s surface area, the analysis was carried out by calculating the difference between the surface area of the reconstructed model and the planar area of the test area. The surface area of the reconstructed model comprises the surface area of the model and the void area within the target site. The void area is substituted with the planar area during the evaluation and analysis of the test outcomes. The surface area of the model is determined using the query function in 3DReshaper, while the void area is assessed using CAD software (version 2020). The 3D model boundary is automatically delineated within 3DReshaper. Subsequently, the gap between these wireframes is computed using the measurement function after importing the boundary wireframe of the test area and the 3D model in DXF format into CAD. Versions 2018 of 3DReshaper and 2020 of AutoCAD were used. For model construction in 3DReshaper, point cloud files in *LAS* format were imported, followed by noise filtering and registration. A triangular mesh was applied for mesh division, and nearest neighbor interpolation filled gaps in the point cloud. Regular sampling used a uniform point distance of 0.05 m, and hole management was applied to generate a complete surface. Surface flatness was evaluated using a scale size of 2 mm, and errors were visualized with color mapping to assess scanning accuracy and optimize the reconstructed model.

Figure 16a shows the variation in surface area between the reconstructed model and the planar area of the test site under different sampling intervals. The surface area decreases as the sampling interval increases, with the difference at a 0.01 m interval approximately ten times greater than at a 0.30 m interval, indicating a strong decreasing trend primarily controlled by the sampling interval. Figure 16b presents statistical results for surface area differences at various station distances. Although the differences are relatively small, a clear trend appears: the surface area decreases as the station distance increases from 5 m to 30 m, representing a reduction of nearly 20%. This indicates that station distance affects surface area, although the effect is less pronounced than that of the sampling interval. Figure 16c summarizes surface area differences at different scanning angles. The variations are generally consistent across angles, suggesting that scanning angle has a minimal direct effect on surface area. However, slight fluctuations are observed at smaller angles, which may result from partial occlusion and reduced overlap of the point cloud in regions with complex topography.

Multi-factor variance analysis using SPSS (Statistics 26) was conducted to evaluate the effects of sampling interval, scanning angle, and station distance on the surface area of the three-dimensional model, as shown in Table 8. The sum of squares represents the total squared deviations from the mean and reflects overall data variability. Degrees of freedom indicate the number of independent values that can vary when estimating population parameters from a sample. The F-value, calculated as the ratio of the mean sum of squares between groups to that within groups, indicates the significance of differences, while the *p*-value measures the level of statistical significance. Differences are considered significant if 0.01 < *p* < 0.05 and highly significant if *p* < 0.01 [41,47]. The analysis shows that sampling interval and station distance have a highly significant effect on the surface area of the three-dimensional model (*p* < 0.01), indicating their critical role in three-dimensional laser scanning accuracy. In contrast, the scanning angle does not significantly affect the surface area (*p* > 0.05). Comparison of the sum of squares reveals that the sampling interval has a greater impact on surface area than station distance.

## 4. Conclusions

This study aims to acquire high-quality three-dimensional scanning data of fill surfaces by investigating the effects of sampling interval, station distance, and scanning angle on point cloud data and reconstructed models through field experiments. The following conclusions were drawn:(1)In three-dimensional laser scanning of fill surfaces, the point cloud exhibits high density near the setup station, gradually becoming sparser with distance. At a scanning angle of 90°, the point cloud also shows left-right symmetry. Statistical analysis reveals that the number of points in the target area decreases exponentially with the sampling interval and logarithmically with increasing station distance. The sampling interval exerts a more pronounced influence on point cloud density compared to station distance and scanning angle. The number of points at *β* = 30° increases by approximately 40% compared to *β* = 90°, with *β* = 60° and 90° showing similar numbers.(2)The three-dimensional model of the fill surface was reconstructed using the Delaunay triangulation algorithm based on point cloud data. Results indicate that both sampling interval and station distance affect the resolution of the reconstructed surface model and its ability to accurately capture bumps and undulations, with resolution decreasing as sampling interval and station distance increase. A larger sampling interval can lead to missing edges in the reconstructed model. The accuracy of the reconstructed surface model shows minimal sensitivity to changes in scanning angle.(3)When conducting three-dimensional laser scanning of fill surfaces with a total station, the setup station should ideally be positioned as close as possible to the target area while meeting minimum scanning distance requirements. If site constraints necessitate a greater station distance, reducing the sampling interval can enhance scanning accuracy. Ideally, the setup station should align with the symmetry line along the longer side of the target area. However, if site conditions prevent this alignment, a moderate deviation from the symmetry line has a negligible impact on scanning outcomes. A sampling interval of 0.03 m is recommended to achieve a balance between scanning accuracy and efficiency.

## Figures and Tables

**Figure 1 sensors-25-07174-f001:**
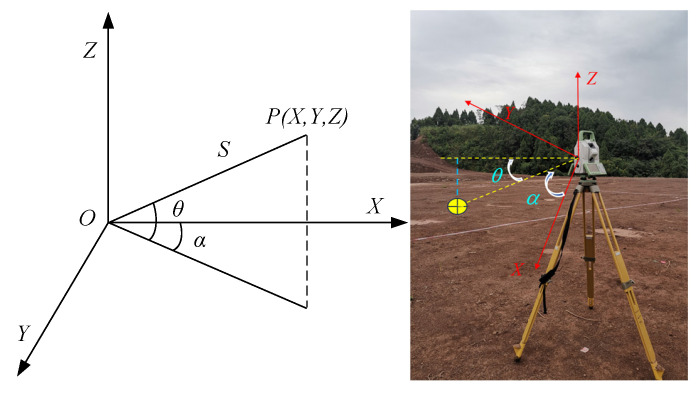
Principle for calculating scanning point coordinates.

**Figure 2 sensors-25-07174-f002:**
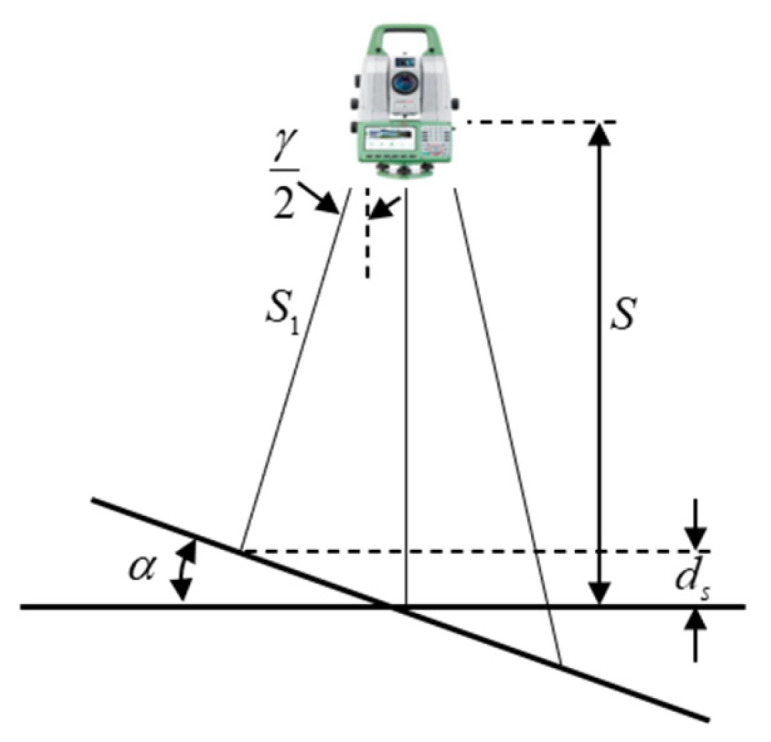
Ranging deviation due to target object tilt.

**Figure 3 sensors-25-07174-f003:**
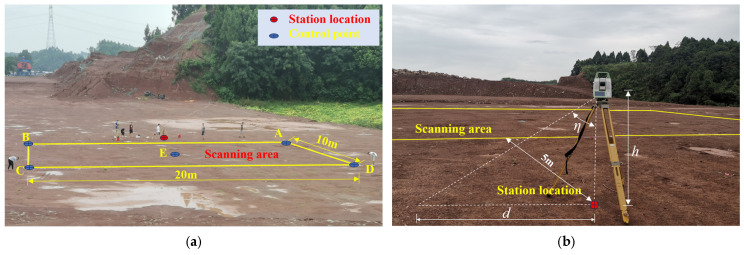
Scanning Area Site Schematic. (**a**) Layout of the scanning area; (**b**) Station arrangement of the scanner.

**Figure 4 sensors-25-07174-f004:**
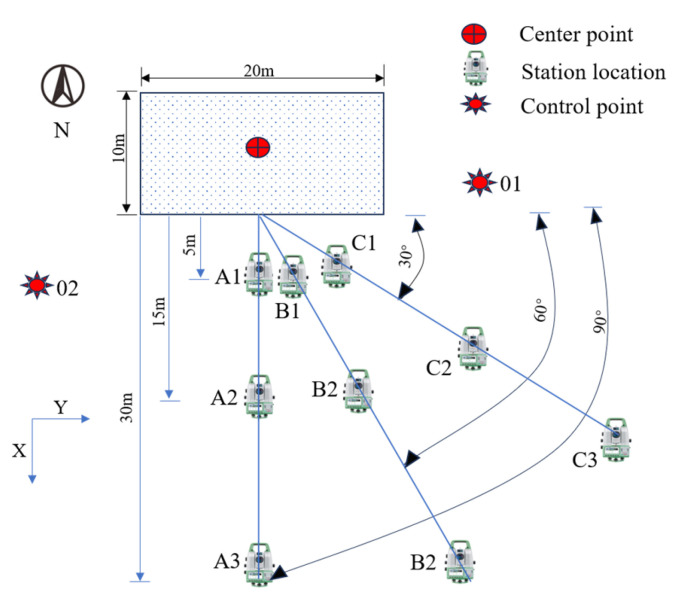
Schematic of Total Station 3D Scanner Station Locations.

**Figure 5 sensors-25-07174-f005:**
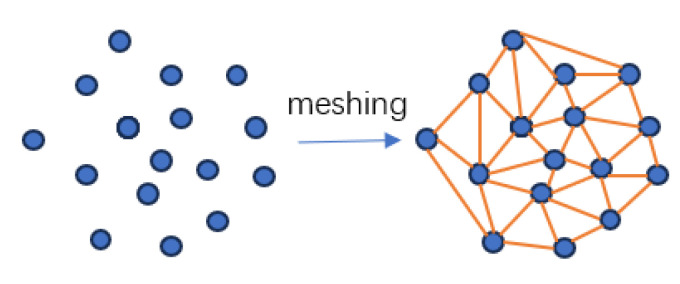
Delaunay Triangular Sectioning.

**Figure 6 sensors-25-07174-f006:**
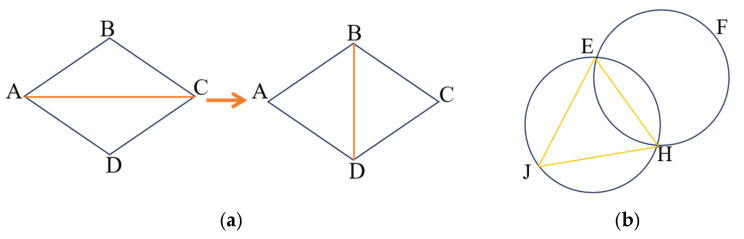
Delaunay Triangulation. (**a**) Maximized minimum angle characteristics; (**b**) External connection characteristics of empty circles.

**Figure 7 sensors-25-07174-f007:**
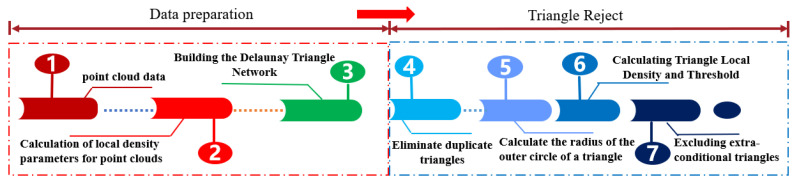
Flow of the Delaunay Triangulation Algorithm.

**Figure 8 sensors-25-07174-f008:**
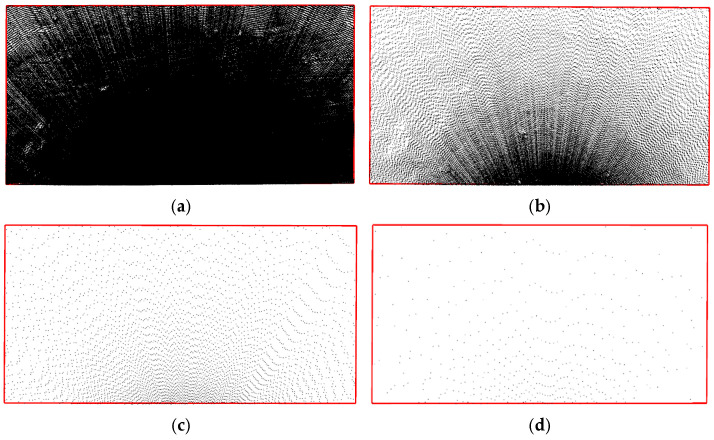
Point Cloud Images with Different Sampling (*L* = 5 m, *β* = 90°). (**a**) *Δ* = 0.01 m; (**b**) *Δ* = 0.03 m; (**c**) *Δ* = 0.10 m; (**d**) *Δ* = 0.30 m.

**Figure 9 sensors-25-07174-f009:**
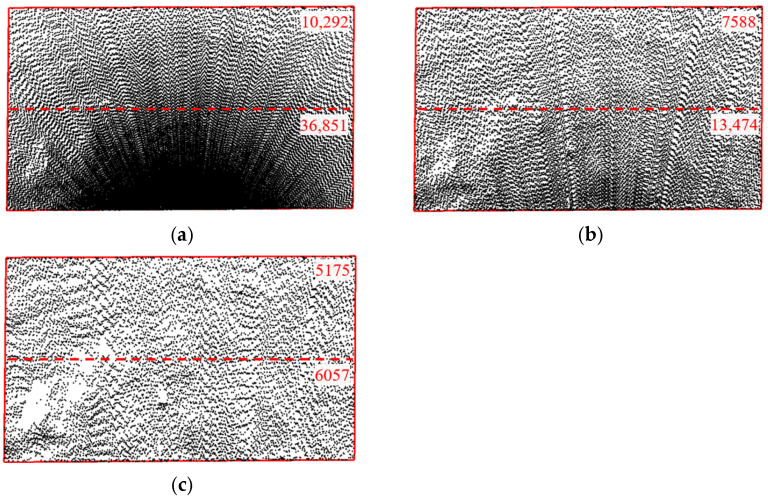
Point Cloud Images with Different Station Distances (*Δ* = 0.03 m, *β* = 90°). (**a**) *L* = 5 m; (**b**) *L* = 10 m; (**c**) *L* = 15 m.

**Figure 10 sensors-25-07174-f010:**
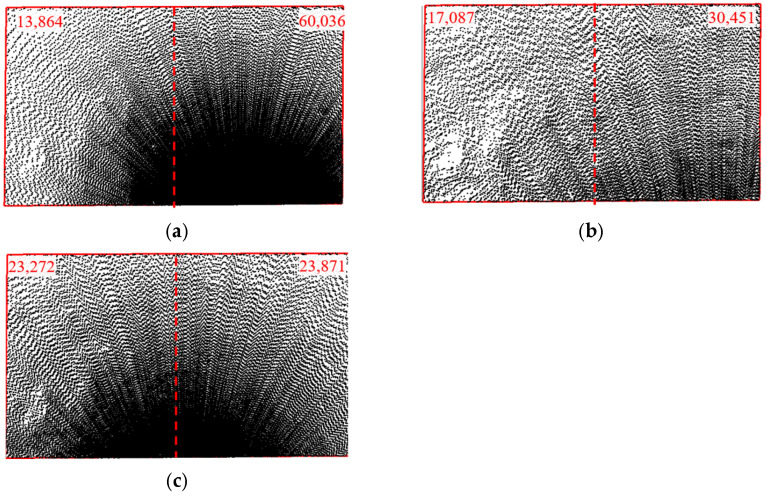
Point Cloud Images with Different Scanning Angles (*Δ* = 0.03 m, *L* = 5 m). (**a**) *β* = 30°; (**b**) *β* = 60°; (**c**) *β* = 90°.

**Figure 11 sensors-25-07174-f011:**
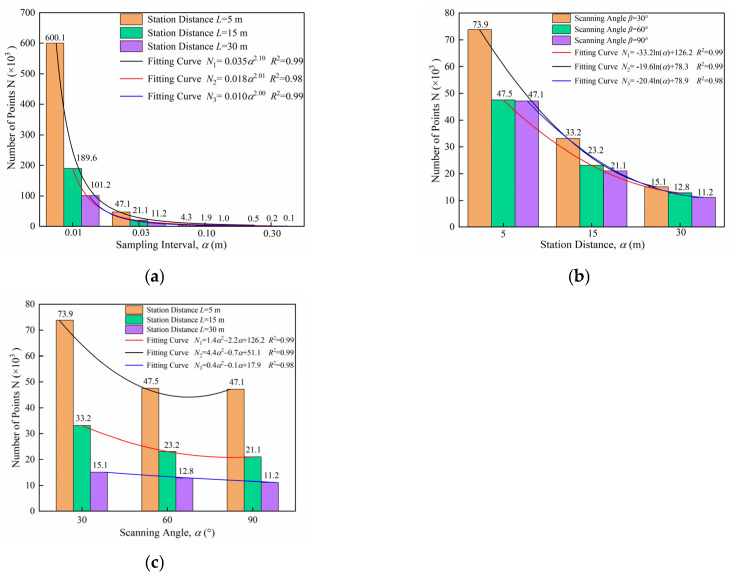
Number of Points under Different Scanning Parameters. (**a**) Variation with sampling interval; (**b**) Variation with station distance; (**c**) Variation with scanning angle.

**Figure 12 sensors-25-07174-f012:**
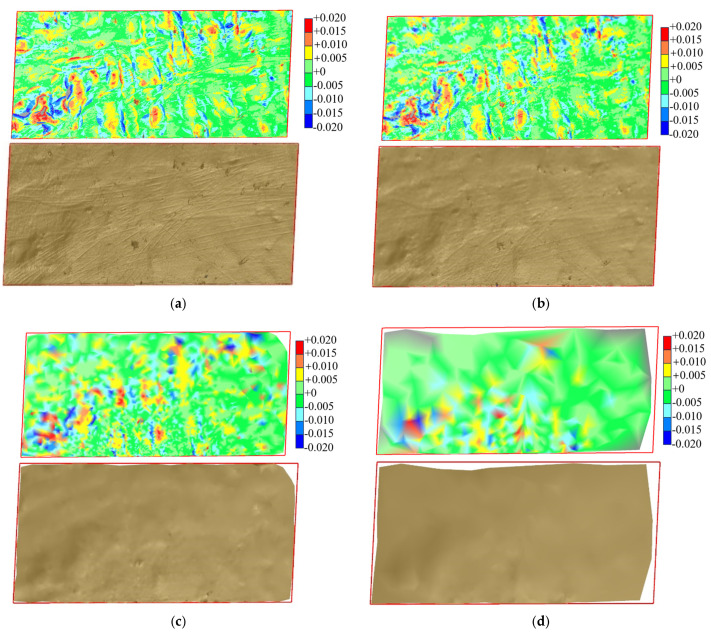
Reconstructed Model of Fill Surface with Different Sampling Intervals (*L* = 5 m, *β* = 90°). (**a**) *Δ* = 0.01 m; (**b**) *Δ* = 0.03 m; (**c**) *Δ* = 0.10 m; (**d**) *Δ* = 0.30 m.

**Figure 13 sensors-25-07174-f013:**
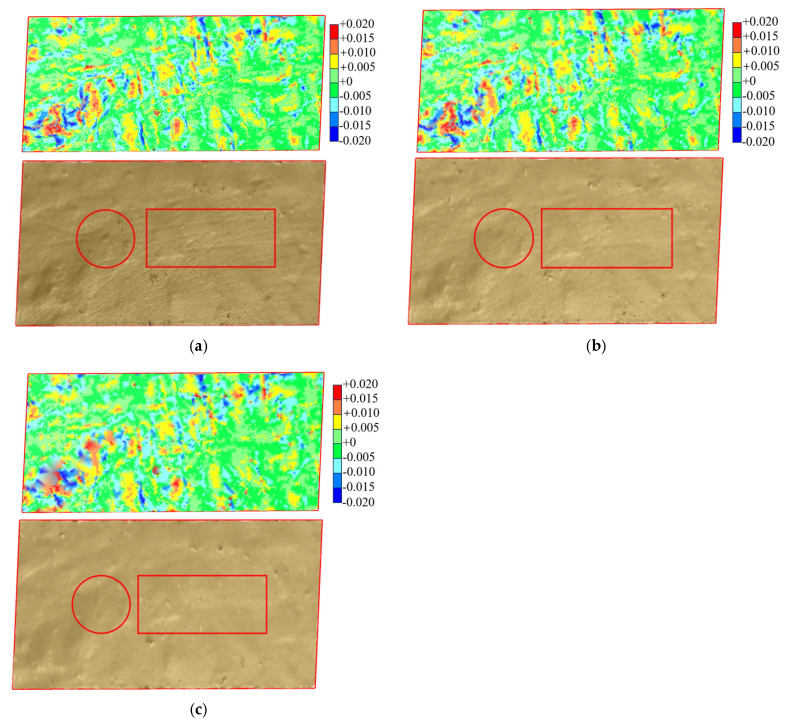
Reconstructed Model of Fill Surface with Different Station Distances (*Δ* = 0.03 m, *β* = 90°). (**a**) *L* = 5 m; (**b**) *L* = 15 m; (**c**) *L* = 30 m.

**Figure 14 sensors-25-07174-f014:**
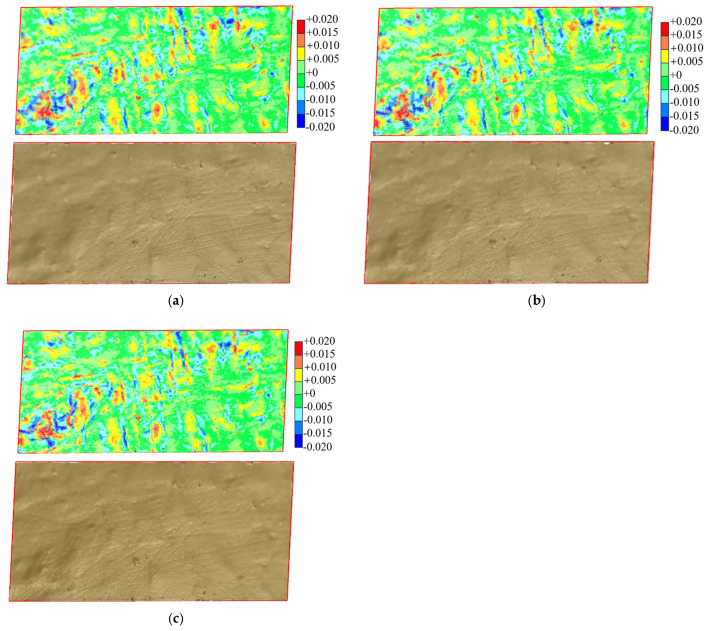
Reconstructed Model of Fill Surface with Different Scanning Angles (*Δ* = 0.03 m, *L* = 5 m). (**a**) *β* = 30°; (**b**) *β* = 60°; (**c**) *β* = 90°.

**Figure 15 sensors-25-07174-f015:**
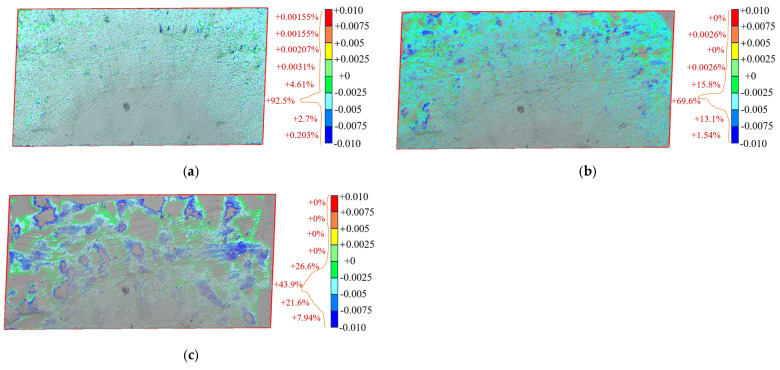
Surface differences in reconstructed fill models at different sampling intervals. (**a**) *Δ* = 0.01 m vs. *Δ* = 0.03 m; (**b**) *Δ* = 0.03 m vs. *Δ* = 0.10 m; (**c**) *Δ* = 0.03 m vs. *Δ* = 0.30 m.

**Figure 16 sensors-25-07174-f016:**
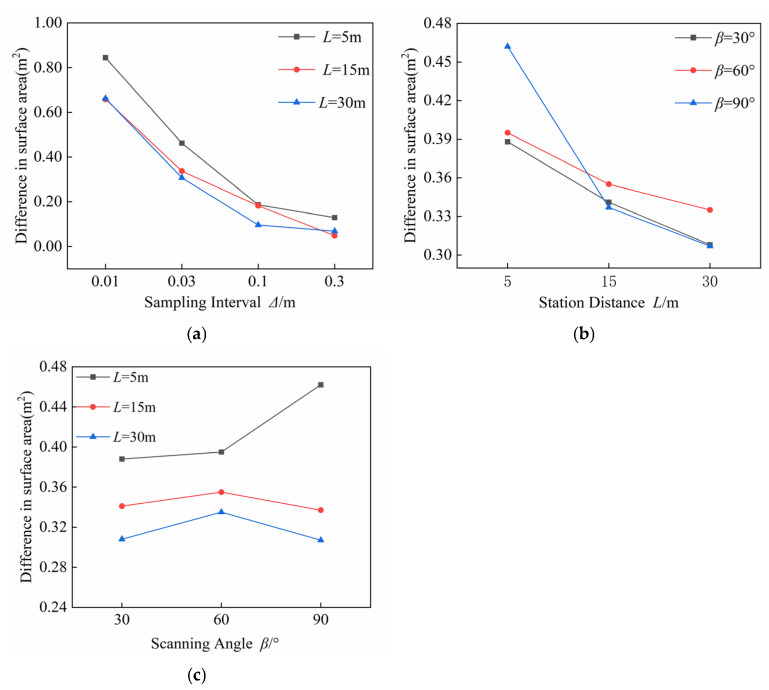
Surface Area Differences in Reconstructed Fill Models under Various Scanning Parameters. (**a**) Different sampling intervals; (**b**) Different station distances; (**c**) Different scanning angles.

**Table 1 sensors-25-07174-t001:** Summary of laser-based techniques for road assessment.

References	Laser-Based Scanning Device	Application Scenarios
[5]	N/A	The deformation of highway and railway embankments
[6,7]	N/A	The detection of road surface damage
[8]	FARO	Extracting concrete runway irregularities, identifying road surface irregularities.
[9]	Leica TC2002	Calculated fault values between concrete slabs using TLS data to assess horizontal plane differences
[10]	Riegl LMS Z-420i	Measure longitudinal cracks in jointed concrete pavements
[11]	MLS	Identify hazards on road surfaces and classify each full-size pavement
[12]	FARO	Scan large concrete areas and assess rigid pavement slab defects

**Table 2 sensors-25-07174-t002:** Overview of laser scanning parameter optimization.

References	Laser-Based Scanning Device	Research Object
[13,14,15,16]	FARO	Color, roughness, incident angle, and distance on point cloud data quality
[17,18]	Leica ScanStation2	Surface roughness, color, shape, and scanning conditions
[19]	N/A	Incident angle, laser range, sampling interval, and station height
[20]	Leica ScanStation P40 and Topcon GLS-1500	Plane residuals, scanning distance, and incident angle.

**Table 3 sensors-25-07174-t003:** Parameters of Scanning Conditions.

Station Location	*Β* (°)	*L* (m)	*Δ* (m)
A1	90	5	0.01, 0.03, 0.10, 0.30
A2		15	
A3		30	
B1	60	5	0.03
B2		15	
B3		30	
C1	30	5	0.03
C2		15	
C3		30	

**Table 4 sensors-25-07174-t004:** Number of Points and Scanning Time for Different Sampling Intervals and Station Distances.

Station Location	*L* (m)	*Δ* (m)	Number of Points	Scanning Time
A1	5	0.01	600,033	21′21″
		0.03	47,143	4′35″
		0.10	4250	1′20″
		0.30	461	0′36″
A2	15	0.01	189,623	12′51″
		0.03	21,062	2′43″
		0.10	1882	0′55″
		0.30	205	0′32″
A3	30	0.01	101,227	8′20″
		0.03	11,232	2′31″
		0.10	1002	0′43″
		0.30	114	0′16″

**Table 5 sensors-25-07174-t005:** Elevation deviation data from ground laser scanning and precise leveling measurements based on the check points (*L* = 15 m and *β* = 90°).

*Δ* (m)	A	B	C	D	E	RMSE
0.01	0.006	0.005	0.008	0.006	0.005	0.0136
0.03	0.007	0.006	0.008	0.008	0.006	0.0149
0.1	0.019	0.018	0.022	0.025	0.015	0.0449
0.3	0.024	0.022	0.026	0.022	0.019	0.0508

**Table 6 sensors-25-07174-t006:** Elevation deviation data derived from ground laser scanning and precise leveling measurements based on the 5 check points (Scanning angle *β* = 90° and Sampling interval *Δ* = 0.03 m).

*L* (m)	A	B	C	D	E	RMSE
5	0.007	0.008	0.013	0.012	0.002	0.0207
15	0.011	0.012	0.02	0.019	0.008	0.0331
30	0.019	0.017	0.024	0.022	0.011	0.0428

**Table 7 sensors-25-07174-t007:** Elevation deviation data derived from ground laser scanning and precise leveling measurements based on the 5 check points (Station distance *L* = 5 m and Sampling interval *Δ* = 0.03 m).

*Β* (°)	A	B	C	D	E	RMSE
30	0.018	0.007	0.012	0.021	0.007	0.0317
60	0.014	0.009	0.017	0.018	0.006	0.0304
90	0.007	0.008	0.013	0.012	0.002	0.0207

**Table 8 sensors-25-07174-t008:** Significant analysis of the influence of different factors on the surface area of the 3D model.

Factor	Type III Sum of Squares	Degrees of Freedom for Statistical Variables (*V_df_*)	Mean Square	Analysis of Variance Statistic (*F*)	Significance (*p*)
Sampling Interval	0.744	3	0.248	212.411	0.002
Station Distance	0.048	2	0.024	20.654	0.003
Scanning Angle	0.001	2	0.001	0.453	0.645

## Data Availability

The original contributions presented in this study are included in the article. Further inquiries can be directed to the corresponding author.

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
