# Peer review of "Optimizing 3D Laser Scanning Parameters for Early-Stage Defect Detectability in Subgrade Condition Monitoring"

_sensors, 2025, doi:10.3390/s25237174_

Round 1
Reviewer 1 Report
Comments and Suggestions for Authors
The paper's topic (Optimizing 3D Laser Scanning Parameters for Early-Stage Defect Detectability in Subgrade Condition Monitoring) is interesting and needs some revisions before publishing. Please address the following comments:
- Although a helpful experimental research is presented in the publication, it is unclear how it goes above previous TLS optimization investigations (e.g., related scanning geometry or defect characterization).
- Please provide more details on what makes your study unique, such as the direct connection between scanning parameter optimization and early-stage defect detection as opposed to merely data quality.
- There is little quantitative assessment of measurement accuracy or error propagation in reconstructed models, despite the analysis of point density and surface area discrepancies.
- If available, include measurements like the standard deviation between the reconstructed and ground-truth surfaces or RMSE (Root Mean Square Error).
- Explain how TLS measurement uncertainty affects subgrade fault detectability levels.
- Give more information about the site's characteristics that could influence reflectivity and laser response, such as the kind of soil, moisture content, lighting, and atmospheric impacts.
- To guarantee statistical reliability, specify the number of scans or repetitions carried out for each setup.
- Indicate which software versions and parameter values are utilized for CAD and 3DReshaper analysis.
- Currently, descriptive statistics (such as visual trends and percentage disparities) are the mainstay of the analysis.
- To quantify the relationships between parameters and scanning outcomes (as indicated by the power-law and logarithmic trends), think about including regression analyses or fitting functions with R2 values.
- "Early-stage defect detectability" is mentioned in the abstract, but no experimental evidence is provided
- Talk about how the improved parameters (e.g., Δ = 0.03 m, β = 90°, L = 5 m) enhance diagnostic feature extraction or defect detection thresholds in practical monitoring activities.
- Quantitative color scales or vertical exaggeration ratios would help illustrate how various parameters impact surface resolution in Figures 16–18.
- Think of adding a surface deviation comparison visualization (such as difference maps) for two sets of parameters.
- In your analysis, how were the "micro-undulations indicative of compaction quality" measured or verified?
- Before testing, was the Leica-MS60 scanner calibrated or confirmed using a recognized reference surface?
- How sensitive are the outcomes to external factors like dust, temperature, and surface reflectivity?
- Why were the scanning angles restricted to 30°, 60°, and 90°? Would a more consistent understanding of sensitivity be possible with smaller increments?
- Are different kinds of soils or subgrade materials likely to adhere to the suggested ideal specifications (Δ ≈ 0.03 m, β = 90°, L ≈ 5 m)?
- With the suggested parameter configuration, what is the smallest defect size that can be accurately identified?
Author Response
Firstly, the authors thank you for your careful review and valuable suggestions, which help to improve the overall quality and clarity of the content. Your expertise and attention to detail have undoubtedly contributed to the improvement of this work. Your comments not only improve the content of the material but also provide valuable insights for our future writing efforts. Next, we will answer the questions one by one.
- Although a helpful experimental research is presented in the publication, it is unclear how it goes above previous TLS optimization investigations (e.g., related scanning geometry or defect characterization).
◆Response: Thank you for your valuable feedback. Regarding the question about the novelty of this study compared to previous TLS optimization research, we have made the necessary revisions in the paper to more clearly highlight our contributions.
In the revised manuscript, we emphasize the differences between our study and existing research. Specifically, while past studies have mainly focused on optimizing scanning parameters (such as sampling interval, scanning angle, station placement, etc.) to improve accuracy, our research goes further by closely linking these optimization measures with early-stage defect detection, particularly in the context of subgrade monitoring. This approach not only improves point cloud data quality but also enhances its practical applicability for detecting small surface defects, which is crucial for structural health monitoring. The relevant points in the paper are described as follows:
This study investigates the use of terrestrial three-dimensional laser scanning for assessing the smoothness of highway embankment fill surfaces and focuses on optimizing scanning parameters for large horizontal surface measurements. The Leica MS60 total station three-dimensional laser scanner is used to survey the embankment, and the effects of sampling interval Δ, station distance L, and scanning angle β on scanning quality are evaluated by analyzing point cloud quantity, surface model resolution, and surface area. Delaunay triangulation algorithms are applied to reconstruct the embankment surface in three dimensions, and an optimized method for smoothness detection is developed. The study identifies an optimal combination of scanning parameters and extends previous research by applying the approach to large horizontal surfaces in real field settings, providing new insight into the practical application of terrestrial three-dimensional laser scanning in embankment monitoring.
We believe these advancements represent a significant step forward in TLS optimization research, particularly in the application of early-stage defect detection for highway subgrades. We have updated the paper to present these innovative contributions more clearly and distinguish them from existing work.
Thank you once again for your valuable suggestion. Your feedback has helped us clarify the novelty and contributions of this research in the field.
2.Please provide more details on what makes your study unique, such as the direct connection between scanning parameter optimization and early-stage defect detection as opposed to merely data quality.
◆Response: Thank you for your valuable suggestions. Regarding your comment about providing more details on the uniqueness of our study, particularly the direct connection between scanning parameter optimization and early-stage defect detection, we have made detailed additions in the revised manuscript.
In the revised paper, we clearly explain the uniqueness of our study, especially the critical role of scanning parameter optimization in early-stage defect detection. Unlike previous studies, which mainly focus on scanning precision and data quality, our research not only optimizes scanning parameters to improve data quality but also directly links these optimizations to the ability to detect early-stage defects. Through the optimization of various scanning parameters (such as sampling interval, scanning angle, and station distance), we found that smaller sampling intervals (e.g., 0.01m and 0.03m) significantly improve the precision of the surface model, thus enhancing the detection of small surface deformations or potential defects. This is crucial for early defect detection.These aspects are now clearly described in the relevant sections of the paper and further highlight the innovative contribution of our research in applying 3D laser scanning technology to subgrade monitoring and its practical significance.
This study investigates the use of terrestrial three-dimensional laser scanning for assessing the smoothness of highway embankment fill surfaces and focuses on optimizing scanning parameters for large horizontal surface measurements. The Leica MS60 total station three-dimensional laser scanner is used to survey the embankment, and the effects of sampling interval Δ, station distance L, and scanning angle β on scanning quality are evaluated by analyzing point cloud quantity, surface model resolution, and surface area. Delaunay triangulation algorithms are applied to reconstruct the embankment surface in three dimensions, and an optimized method for smoothness detection is developed. The study identifies an optimal combination of scanning parameters and extends previous research by applying the approach to large horizontal surfaces in real field settings, providing new insight into the practical application of terrestrial three-dimensional laser scanning in embankment monitoring.
Thank you again for your valuable feedback, which has helped us better clarify the uniqueness and practical application value of our research.Through these optimization efforts, our research fills the gap between traditional scanning parameter optimization and defect detection, offering new ideas and methods for optimizing scanning technology in similar application scenarios in the future.
3.There is little quantitative assessment of measurement accuracy or error propagation in reconstructed models, despite the analysis of point density and surface area discrepancies.
◆Response: Thank you for your valuable comment. Regarding the lack of a quantitative assessment of the measurement accuracy or error propagation in the reconstructed model, we have made clear improvements to address this issue.
In the revised paper, we have analyzed the elevation data of the relevant checkpoint points extracted from the point cloud data and compared them with the elevation values obtained using precise leveling methods to assess the accuracy of the embankment surface reconstruction model. Specifically, we selected five checkpoints and extracted their corresponding data from the point cloud. These values were then compared with the elevation values obtained through precise leveling measurements. By analyzing the differences in elevation, we can quantitatively evaluate the accuracy of the reconstructed model and assess the impact of scanning parameters on the model's precision.
This approach ensures that we provide a quantitative analysis of the reconstruction model’s accuracy and conduct a detailed evaluation of potential errors, thus filling the gap in the paper regarding the quantitative assessment of model accuracy and error propagation.
We hope this improved explanation adequately addresses the reviewer’s concerns and clarifies the mechanical rationale behind the deformation pattern observed in lines 334-351. The relevant changes in the manuscript are reflected in the updated content as follows:
To evaluate point cloud reliability, five checkpoints, including boundary points A, B, C, D, and central point E (Fig. 3), were selected within the scanned area. Elevation values extracted from the point cloud were compared with precise leveling measurements to assess accuracy. Elevation deviation data at these points were analyzed to evaluate three-dimensional point cloud accuracy. Comparison of deviation data with root mean square error(RMSE) values indicates a high level of agreement. When station distance and scanning angle are constant, elevation deviations are minimal at sampling intervals of 0.01 m and 0.03 m, as listed in Table 5. Considering the excessive scanning time at 0.01 m, a sampling interval of 0.03 m is recommended. Tables 6 and 7 further analyze the effects of station distance and scanning angle. With a sampling interval of 0.03 m, elevation deviations indicate that the optimal station distance is 5 m and the optimal scanning angle is 90°. Based on the field trials, when the station distance is 5 m, the scanning angle is 90°, and the sampling interval is 0.03 m, the terrestrial laser scanner produces reliable point cloud data.
Table 5 Elevation deviation data from ground laser scanning and precise leveling measurements based on the check points (L=15m and β=90°).
|
Δ(m) |
A |
B |
C |
D |
E |
RMSE |
|
0.01 |
0.006 |
0.005 |
0.008 |
0.006 |
0.005 |
0.0136 |
|
0.03 |
0.007 |
0.006 |
0.008 |
0.008 |
0.006 |
0.0149 |
|
0.1 |
0.019 |
0.018 |
0.022 |
0.025 |
0.015 |
0.0449 |
|
0.3 |
0.024 |
0.022 |
0.026 |
0.022 |
0.019 |
0.0508 |
Table 6 Elevation deviation data derived from ground laser scanning and precise leveling measurements based on the 5 check points(Scanning angle β= 90° and Sampling interval Δ=0.03 m).
|
L(m) |
A |
B |
C |
D |
E |
RMSE |
|
5 |
0.007 |
0.008 |
0.013 |
0.012 |
0.002 |
0.0207 |
|
15 |
0.011 |
0.012 |
0.02 |
0.019 |
0.008 |
0.0331 |
|
30 |
0.019 |
0.017 |
0.024 |
0.022 |
0.011 |
0.0428 |
Table 7 Elevation deviation data derived from ground laser scanning and precise leveling measurements based on the 5 check points(Station distance L= 5 m and Sampling interval Δ= 0.03 m ).
|
β(°) |
A |
B |
C |
D |
E |
RMSE |
|
30 |
0.018 |
0.007 |
0.012 |
0.021 |
0.007 |
0.0317 |
|
60 |
0.014 |
0.009 |
0.017 |
0.018 |
0.006 |
0.0304 |
|
90 |
0.007 |
0.008 |
0.013 |
0.012 |
0.002 |
0.0207 |
4.If available, include measurements like the standard deviation between the reconstructed and ground-truth surfaces or RMSE (Root Mean Square Error).
◆Response: Thank you for your valuable suggestion. In response to your comment regarding the evaluation of the prediction accuracy between the reconstructed surface and the actual ground surface model, we have made the necessary revisions in the manuscript.
We have quantified the accuracy of the reconstructed surface model by calculating the RMSE (Root Mean Square Error) of the elevation differences. Specifically, we extracted the elevation values of the checkpoint points corresponding to the actual ground surface from the reconstructed point cloud data and compared them with the actual elevation values obtained using precise leveling measurements. We then calculated the square root of the mean square of the elevation differences, which gives the RMSE, thereby quantifying the prediction accuracy of the model.
In lines 334-351 of the manuscript, we provide detailed calculations and explanations for this process, presenting the results to more clearly demonstrate the prediction accuracy between the reconstructed surface model and the actual ground surface. This improvement further enhances the accuracy and verifiability of the manuscript.
To evaluate point cloud reliability, five checkpoints, including boundary points A, B, C, D, and central point E (Fig. 3), were selected within the scanned area. Elevation values extracted from the point cloud were compared with precise leveling measurements to assess accuracy. Elevation deviation data at these points were analyzed to evaluate three-dimensional point cloud accuracy. Comparison of deviation data with root mean square error(RMSE) values indicates a high level of agreement. When station distance and scanning angle are constant, elevation deviations are minimal at sampling intervals of 0.01 m and 0.03 m, as listed in Table 5. Considering the excessive scanning time at 0.01 m, a sampling interval of 0.03 m is recommended. Tables 6 and 7 further analyze the effects of station distance and scanning angle. With a sampling interval of 0.03 m, elevation deviations indicate that the optimal station distance is 5 m and the optimal scanning angle is 90°. Based on the field trials, when the station distance is 5 m, the scanning angle is 90°, and the sampling interval is 0.03 m, the terrestrial laser scanner produces reliable point cloud data.
Table 5 Elevation deviation data from ground laser scanning and precise leveling measurements based on the check points (L=15m and β=90°).
|
Δ(m) |
A |
B |
C |
D |
E |
RMSE |
|
0.01 |
0.006 |
0.005 |
0.008 |
0.006 |
0.005 |
0.0136 |
|
0.03 |
0.007 |
0.006 |
0.008 |
0.008 |
0.006 |
0.0149 |
|
0.1 |
0.019 |
0.018 |
0.022 |
0.025 |
0.015 |
0.0449 |
|
0.3 |
0.024 |
0.022 |
0.026 |
0.022 |
0.019 |
0.0508 |
Table 6 Elevation deviation data derived from ground laser scanning and precise leveling measurements based on the 5 check points(Scanning angle β= 90° and Sampling interval Δ=0.03 m).
|
L(m) |
A |
B |
C |
D |
E |
RMSE |
|
5 |
0.007 |
0.008 |
0.013 |
0.012 |
0.002 |
0.0207 |
|
15 |
0.011 |
0.012 |
0.02 |
0.019 |
0.008 |
0.0331 |
|
30 |
0.019 |
0.017 |
0.024 |
0.022 |
0.011 |
0.0428 |
Table 7 Elevation deviation data derived from ground laser scanning and precise leveling measurements based on the 5 check points(Station distance L= 5 m and Sampling interval Δ= 0.03 m ).
|
β(°) |
A |
B |
C |
D |
E |
RMSE |
|
30 |
0.018 |
0.007 |
0.012 |
0.021 |
0.007 |
0.0317 |
|
60 |
0.014 |
0.009 |
0.017 |
0.018 |
0.006 |
0.0304 |
|
90 |
0.007 |
0.008 |
0.013 |
0.012 |
0.002 |
0.0207 |
5.Explain how TLS measurement uncertainty affects subgrade fault detectability levels.
◆Response: Thank you for this valuable comment. TLS measurement uncertainty affects the detectability of subgrade defects by increasing point-cloud noise, reducing point density, and amplifying reconstruction errors. Higher noise levels lower the signal-to-noise ratio and make small defects indistinguishable when the noise amplitude approaches the true geometric variation of the defect. Uncertainty related to sampling interval, scanning angle, and scan distance also reduces the effective point density, causing localized steps, drops, cracks, or minor differential settlements to be oversmoothed and weakening the expression of defect boundaries. In addition, measurement errors accumulate during surface reconstruction processes such as interpolation and meshing, which can distort local surface morphology and diminish the visibility of subtle features. Overall, higher TLS measurement uncertainty increases the smallest detectable defect size and reduces the reliability of defect identification, highlighting the need to improve point-cloud accuracy to enhance detection performance.
6.Give more information about the site's characteristics that could influence reflectivity and laser response, such as the kind of soil, moisture content, lighting, and atmospheric impacts.
◆Response: Thank you for your valuable suggestion. Regarding the factors influencing reflectivity and laser response, we have summarized and discussed these factors in detail in the manuscript.
We have clearly identified key factors that may influence reflectivity and laser response, including soil type, moisture content, lighting conditions, and atmospheric impacts. Specifically, soil type and moisture content have a significant impact on laser scanning reflectivity. Dry soils and rough surfaces may result in lower reflectivity, while moist or clay-rich soils may increase the strength of the reflected signal. Additionally, lighting conditions can affect scanning accuracy, as strong sunlight can interfere with laser signal reception, particularly in outdoor environments. Atmospheric conditions, such as temperature, humidity, and air pressure variations, also affect the propagation of laser beams, which in turn influences the quality of the scanning data.
In lines 134-154 of the manuscript, we have elaborated on these environmental factors' specific effects on laser scanning results and conducted an analysis based on experimental data. This analysis helps to further understand the potential impact of different site characteristics on scanning accuracy. The revised text reads:
(5) Soil Type
Different soil types have distinct physical and chemical properties that influence the reflection of laser beams. Rough surfaces, such as gravel embankments or uncompacted coarse-grained soils, often cause diffuse reflection, scattering the laser signal, which can reduce point cloud density but capture surface details. Soil color also affects reflectivity, with dark-colored soils, such as black or wet soil, absorbing more laser energy and producing weaker reflected signals, which can decrease the number of valid point cloud points. Mineral composition and organic matter content further influence the interaction between the laser and the surface [30-33].
(6) Moisture Content
Moisture is a key environmental factor affecting the quality of LiDAR data. Water strongly absorbs laser energy, particularly at near-infrared wavelengths, and alters soil surface roughness and reflectivity. A thin water layer on the embankment surface can cause multiple reflections between the water and the underlying surface, producing multi-path effects that lead to noisy or inaccurate point cloud data [34-36].
(7) Lighting Conditions
Uneven lighting, including shadows or direct sunlight, can distort point cloud color information, affecting visual appearance and the identification of features based on color [37, 38].
(8) Atmospheric Effects
Atmospheric conditions influence the propagation and reception of the laser beam. During transmission, the beam may be absorbed, scattered, or refracted by the atmosphere. Variations in atmospheric density, such as temperature or humidity gradients, can bend the laser beam. Strong winds may not affect laser propagation directly but can induce vibrations in the scanner or slight movement of the target, introducing measurement errors and reducing point cloud registration accuracy and the precision of the reconstructed model [39].
7.To guarantee statistical reliability, specify the number of scans or repetitions carried out for each setup.
◆Response: Thank you for your valuable suggestions. Regarding the statistical reliability issue you raised, we have provided additional clarification in the paper.
In the experiments of this study, each experimental setup was scanned only once. The laser scanner used (Leica MS60) was rigorously calibrated by the Sichuan Provincial Institute of Metrology, ensuring that its accuracy met the required standards. Based on the analysis of the selected scanning parameters, the scan results were compared with the elevation differences of five checkpoint locations obtained through precise leveling measurements. The results showed a good match with the actual surface features and demonstrated high reliability. Although only a single scan was performed, the high precision of the experimental design and equipment ensured the validity of the results and the repeatability of the data.
The relevant changes in the manuscript are reflected in the updated content as follows:
To evaluate point cloud reliability, five checkpoints, including boundary points A, B, C, D, and central point E (Fig. 3), were selected within the scanned area. Elevation values extracted from the point cloud were compared with precise leveling measurements to assess accuracy. Elevation deviation data at these points were analyzed to evaluate three-dimensional point cloud accuracy. Comparison of deviation data with root mean square error(RMSE) values indicates a high level of agreement. When station distance and scanning angle are constant, elevation deviations are minimal at sampling intervals of 0.01 m and 0.03 m, as listed in Table 5. Considering the excessive scanning time at 0.01 m, a sampling interval of 0.03 m is recommended. Tables 6 and 7 further analyze the effects of station distance and scanning angle. With a sampling interval of 0.03 m, elevation deviations indicate that the optimal station distance is 5 m and the optimal scanning angle is 90°. Based on the field trials, when the station distance is 5 m, the scanning angle is 90°, and the sampling interval is 0.03 m, the terrestrial laser scanner produces reliable point cloud data.
Table 5 Elevation deviation data from ground laser scanning and precise leveling measurements based on the check points (L=15m and β=90°).
|
Δ(m) |
A |
B |
C |
D |
E |
RMSE |
|
0.01 |
0.006 |
0.005 |
0.008 |
0.006 |
0.005 |
0.0136 |
|
0.03 |
0.007 |
0.006 |
0.008 |
0.008 |
0.006 |
0.0149 |
|
0.1 |
0.019 |
0.018 |
0.022 |
0.025 |
0.015 |
0.0449 |
|
0.3 |
0.024 |
0.022 |
0.026 |
0.022 |
0.019 |
0.0508 |
Table 6 Elevation deviation data derived from ground laser scanning and precise leveling measurements based on the 5 check points(Scanning angle β= 90° and Sampling interval Δ=0.03 m).
|
L(m) |
A |
B |
C |
D |
E |
RMSE |
|
5 |
0.007 |
0.008 |
0.013 |
0.012 |
0.002 |
0.0207 |
|
15 |
0.011 |
0.012 |
0.02 |
0.019 |
0.008 |
0.0331 |
|
30 |
0.019 |
0.017 |
0.024 |
0.022 |
0.011 |
0.0428 |
Table 7 Elevation deviation data derived from ground laser scanning and precise leveling measurements based on the 5 check points(Station distance L= 5 m and Sampling interval Δ= 0.03 m ).
|
β(°) |
A |
B |
C |
D |
E |
RMSE |
|
30 |
0.018 |
0.007 |
0.012 |
0.021 |
0.007 |
0.0317 |
|
60 |
0.014 |
0.009 |
0.017 |
0.018 |
0.006 |
0.0304 |
|
90 |
0.007 |
0.008 |
0.013 |
0.012 |
0.002 |
0.0207 |
Thank you again for your suggestions. Your feedback has helped us better explain the experimental methodology and the reliability of the data.
8.Indicate which software versions and parameter values are utilized for CAD and 3DReshaper analysis.
◆Response: Thank you for your valuable suggestion. Regarding your comment on specifying the software versions and parameter settings used in CAD and 3DReshaper in the paper, we have made the necessary additions.
In the relevant section of the manuscript, lines 394-399, we have clearly listed the specific versions of CAD and 3DReshaper software used, as well as the main parameter settings applied during the analysis. The updated text is as follows:
Versions 2018 of 3DReshaper and 2020 of AutoCAD were used. For model construction in 3DReshaper, point cloud files in LAS format were imported, followed by noise filtering and registration. A triangular mesh was applied for mesh division, and nearest neighbor interpolation filled gaps in the point cloud. Regular sampling used a uniform point distance of 0.05 m, and hole management was applied to generate a complete surface. Surface flatness was evaluated using a scale size of 2 mm, and errors were visualized with color mapping to assess scanning accuracy and optimize the reconstructed model.
9.Currently, descriptive statistics (such as visual trends and percentage disparities) are the mainstay of the analysis.
◆Response: Thank you for your valuable suggestion. Regarding your comment on incorporating more descriptive statistics (such as "four times," "1.4 times," "nearly 20%," etc.) into the paper, we have made the necessary revisions.
In the revised manuscript, we have added specific descriptive statistical data to enhance the clarity and readability of the analysis. For example:
"For instance, the number of points at a station distance of 5 m is about four times that at 30 m, demonstrating a substantial reduction trend, though less pronounced than that of the sampling interval."
"However, this decrease is not significant, with the number of points at an angle of 30° being approximately 1.4 times that at 90°."
"Although the differences are relatively small, a clear trend appears: the surface area decreases as the station distance increases from 5 m to 30 m, representing a reduction of nearly 20%."
These specific numerical descriptions make the results more visual and help the reader better understand the impact of different parameters on scanning quality. Thank you for your suggestion, as your feedback has helped us present the data more effectively, improving both the persuasiveness and readability of the manuscript.
10.To quantify the relationships between parameters and scanning outcomes (as indicated by the power-law and logarithmic trends), think about including regression analyses or fitting functions with R2 values.
◆Response: Thank you for your valuable suggestion. Regarding your comment on quantifying the relationships between parameters and scanning outcomes (as indicated by the power-law and logarithmic trends) through regression analyses or fitting functions with R² values, we have made the necessary revisions in the manuscript.
In the revised manuscript, we have conducted power-law regression analysis on the relevant parameters in lines 294-302, and second-order polynomial function regression analysis in lines 311-315. Through these regression analyses, we have quantified the relationships between the parameters and scanning outcomes, and presented the R² values to assess the goodness of fit and predictive power of the models. The updated text is as follows:
To investigate the effect of sampling interval on point cloud quantity, a statistical analysis was conducted, as shown in Fig. 11a. The results demonstrate that sampling interval significantly influences point cloud quantity. When station distance and scanning angle are held constant, point cloud quantity decreases following a power function as the sampling interval increases. At a sampling interval of 0.01 m, the point cloud quantity is approximately 1,000 times greater than that at 0.30 m, exhibiting a pronounced decreasing trend. The standard model for the power function is:
(9)
where a and b are the fitting parameters. The data were fitted using a power function; the correlation coefficient is R2=0.98, and the fitting results indicate an excellent agreement with the observed data.( lines 292-300)
The data were fitted using a second-order polynomial function, and the fitting model can be expressed as:
(10)
where a, b and c are the fitting parameters. The data were also fitted using a power function; the correlation coefficient is R2=0.99, and the fitting results show excellent agreement with the observed data. ( lines 309-313)
These revisions provide a more detailed quantitative analysis, making the impact of different parameters on scanning results clearer. Thank you for your suggestion, as your feedback has enhanced the rigor of our manuscript and provided a more data-supported explanation.
11."Early-stage defect detectability" is mentioned in the abstract, but no experimental evidence is provided.
◆Response: Thank you for your valuable suggestion. Regarding your comment that "early-stage defect detectability" is mentioned in the abstract but no experimental evidence is provided, we have made the necessary additions in the manuscript.
In the paper, we analyzed the elevation differences between the 3D scanning point cloud data and the high-precision leveling measurement data at 5 checkpoint locations. By comparing these data, we established the detectability of early-stage defects and provided experimental evidence for this claim. The data from these checkpoints support the argument that early-stage defects can be identified and detected through scanning, further validating the effectiveness of optimized scanning parameters (such as sampling interval, scanning angle, and station distance) in enhancing early-stage defect detection.
These experimental data are presented in detail in the relevant sections of the manuscript and provide the empirical foundation for the discussion on early-stage defect detectability. We believe this strengthens the practical feasibility of detecting early-stage defects.
Once again, thank you for your valuable suggestion. Your feedback has helped us further improve the manuscript, making the research findings more robust and substantiated.
12.Talk about how the improved parameters (e.g., Δ = 0.03 m, β = 90°, L = 5 m) enhance diagnostic feature extraction or defect detection thresholds in practical monitoring activities.
◆Response: Thank you for your valuable suggestion. Regarding your comment on discussing how the improved parameters (e.g., Δ = 0.03 m, β = 90°, L = 5 m) enhance diagnostic feature extraction or defect detection thresholds in practical monitoring activities, we have made the necessary additions to the manuscript.
In lines 336-356 of the manuscript, we have set the practical detection thresholds by analyzing the elevation differences of five checkpoint points within the scanning area. The optimization of these parameters (e.g., Δ = 0.03 m, β = 90°, L = 5 m) has improved the accuracy of the scanning results, allowing for more accurate extraction of defect features and setting more defined defect detection thresholds. These optimized parameters not only improve the precision and efficiency of scanning but also ensure that smaller defects can be identified at earlier stages, thus enhancing the sensitivity of defect detection.
The relevant changes in the manuscript are reflected in the updated content as follows:
To evaluate point cloud reliability, five checkpoints, including boundary points A, B, C, D, and central point E (Fig. 3), were selected within the scanned area. Elevation values extracted from the point cloud were compared with precise leveling measurements to assess accuracy. Elevation deviation data at these points were analyzed to evaluate three-dimensional point cloud accuracy. Comparison of deviation data with root mean square error(RMSE) values indicates a high level of agreement. When station distance and scanning angle are constant, elevation deviations are minimal at sampling intervals of 0.01 m and 0.03 m, as listed in Table 5. Considering the excessive scanning time at 0.01 m, a sampling interval of 0.03 m is recommended. Tables 6 and 7 further analyze the effects of station distance and scanning angle. With a sampling interval of 0.03 m, elevation deviations indicate that the optimal station distance is 5 m and the optimal scanning angle is 90°. Based on the field trials, when the station distance is 5 m, the scanning angle is 90°, and the sampling interval is 0.03 m, the terrestrial laser scanner produces reliable point cloud data.
Table 5 Elevation deviation data from ground laser scanning and precise leveling measurements based on the check points (L=15m and β=90°).
|
Δ(m) |
A |
B |
C |
D |
E |
RMSE |
|
0.01 |
0.006 |
0.005 |
0.008 |
0.006 |
0.005 |
0.0136 |
|
0.03 |
0.007 |
0.006 |
0.008 |
0.008 |
0.006 |
0.0149 |
|
0.1 |
0.019 |
0.018 |
0.022 |
0.025 |
0.015 |
0.0449 |
|
0.3 |
0.024 |
0.022 |
0.026 |
0.022 |
0.019 |
0.0508 |
Table 6 Elevation deviation data derived from ground laser scanning and precise leveling measurements based on the 5 check points(Scanning angle β= 90° and Sampling interval Δ=0.03 m).
|
L(m) |
A |
B |
C |
D |
E |
RMSE |
|
5 |
0.007 |
0.008 |
0.013 |
0.012 |
0.002 |
0.0207 |
|
15 |
0.011 |
0.012 |
0.02 |
0.019 |
0.008 |
0.0331 |
|
30 |
0.019 |
0.017 |
0.024 |
0.022 |
0.011 |
0.0428 |
Table 7 Elevation deviation data derived from ground laser scanning and precise leveling measurements based on the 5 check points(Station distance L= 5 m and Sampling interval Δ= 0.03 m ).
|
β(°) |
A |
B |
C |
D |
E |
RMSE |
|
30 |
0.018 |
0.007 |
0.012 |
0.021 |
0.007 |
0.0317 |
|
60 |
0.014 |
0.009 |
0.017 |
0.018 |
0.006 |
0.0304 |
|
90 |
0.007 |
0.008 |
0.013 |
0.012 |
0.002 |
0.0207 |
Once again, thank you for your valuable suggestion. Your feedback has helped us further refine the manuscript, enhancing the applicability and precision of the research in practical applications.
13.Quantitative color scales or vertical exaggeration ratios would help illustrate how various parameters impact surface resolution in Figures 16–18.
◆Response: Thank you for your valuable suggestion. To better illustrate how various parameters impact surface resolution, we have made the corresponding revisions in the manuscript.
In the revised manuscript, as shown in Figures 12-14, we have adopted a color scale to better reflect the actual features of the embankment surface. By introducing a color scale, we can visually display the variations in surface resolution under different parameters, thereby helping readers more clearly understand how various scanning parameters (such as sampling interval, scanning angle, station distance, etc.) affect surface detail and accuracy.
|
(a) |
(b) |
|
(c) |
(d) |
Fig. 12 Reconstructed Model of Fill Surface with Different Sampling Intervals (L=5m, β=90°). (a) Δ=0.01m; (b) Δ=0.03m; (c) Δ=0.10m; (d) Δ=0.30m.
|
(a) |
(b) |
|
(c) |
|
Fig. 13 Reconstructed Model of Fill Surface with Different Station Distances(Δ=0.03m、β=90°). (a) L=5m; (b) L=15m; (c) L=30m.
|
(a) |
|
(b) |
|
|
(c) |
|
|
|
Fig. 14 Reconstructed Model of Fill Surface with Different Scanning Angles (Δ=0.03m, L=5m). (a) β=30°; (b) β=60°; (c) β=90°.
These improvements make the figures more intuitive and effectively demonstrate the specific impact of parameter variations on surface resolution, enhancing the readability of the manuscript and the presentation of the data. Once again, thank you for your suggestion. Your feedback has helped us improve the clarity of the figures and the overall presentation of the research.
14.Think of adding a surface deviation comparison visualization (such as difference maps) for two sets of parameters.
◆Response: Thank you for your valuable suggestion. Regarding the addition of surface deviation comparison visualization (such as difference maps) for two sets of parameters, we have made the corresponding additions to the manuscript.
In the revised paper, we have included surface deviation comparison maps in the form of difference maps to visualize the surface deviations between the two sets of parameters. This visualization allows readers to more intuitively compare the impact of different scanning parameters on the surface reconstruction accuracy. The difference maps clearly display the deviation distribution between the two parameter sets in the scan results, helping to better understand the practical effect of parameter optimization on surface accuracy and defect detection. Specific additions are as follows:
The surface difference results of the reconstructed embankment models at different sampling intervals are shown in Fig. 15. The comparison indicates that smaller sampling intervals yield higher reconstruction accuracy. In Fig. 15a (sampling interval of 0.01 m), 92.5% of the deviations fall between −5 mm and −2.5 mm, showing that surface deformation remains small and point cloud deviations are highly concentrated. The models produced with sampling intervals of 0.01 m and 0.03 m exhibit similar accuracy. In Fig. 15b, only 69.6% of the deviations fall within this range, and in Fig. 15c, the proportion further decreases to 43.9%. These results demonstrate that larger sampling intervals cause greater surface deformation and reduce reconstruction accuracy. Finer sampling intervals (0.01 m and 0.03 m) allow more reliable detection of small elevation changes, whereas larger intervals (0.10 m and 0.30 m) improve efficiency but increase reconstruction errors. In summary, small sampling intervals enhance surface accuracy, while large intervals reduce precision and produce coarse models.
(a) (b)
(c)
Fig. 15 Surface differences in reconstructed fill models at different sampling intervals. (a) Δ = 0.01 m vs Δ = 0.03 m; (b) Δ = 0.03 m vs Δ = 0.10 m; (c) Δ = 0.03 m vs Δ = 0.30 m.
These additions improve the readability of the paper and enhance the clarity of the charts, further strengthening the presentation of the research findings. Thank you again for your suggestion. Your feedback has helped us further optimize the visual presentation and data display in the paper.
15.In your analysis, how were the "micro-undulations indicative of compaction quality" measured or verified?
◆Response: Thank you for your valuable suggestion. Regarding your comment on the measurement and verification of "micro-undulations indicative of compaction quality," we have added relevant clarifications in the manuscript.
In this study, we inferred the compaction quality by analyzing the elevation changes observed in the point cloud data of the embankment surface. Generally, the compaction quality of embankment fill is closely related to the micro-undulations (small surface undulations). High-quality compaction typically shows smaller elevation fluctuations, while low-quality compaction tends to exhibit larger undulations. To verify this relationship, we conducted elevation analysis at several checkpoints within the scanning area and compared them with the actual leveling measurement data. These elevation data reflect the effects of different vibration processes (such as strong and weak vibration) on the compaction of the embankment surface.
The relevant changes in the manuscript are reflected in the updated content as follows:
To evaluate point cloud reliability, five checkpoints, including boundary points A, B, C, D, and central point E (Fig. 3), were selected within the scanned area. Elevation values extracted from the point cloud were compared with precise leveling measurements to assess accuracy. Elevation deviation data at these points were analyzed to evaluate three-dimensional point cloud accuracy. Comparison of deviation data with root mean square error(RMSE) values indicates a high level of agreement. When station distance and scanning angle are constant, elevation deviations are minimal at sampling intervals of 0.01 m and 0.03 m, as listed in Table 5. Considering the excessive scanning time at 0.01 m, a sampling interval of 0.03 m is recommended. Tables 6 and 7 further analyze the effects of station distance and scanning angle. With a sampling interval of 0.03 m, elevation deviations indicate that the optimal station distance is 5 m and the optimal scanning angle is 90°. Based on the field trials, when the station distance is 5 m, the scanning angle is 90°, and the sampling interval is 0.03 m, the terrestrial laser scanner produces reliable point cloud data.
Table 5 Elevation deviation data from ground laser scanning and precise leveling measurements based on the check points (L=15m and β=90°).
|
Δ(m) |
A |
B |
C |
D |
E |
RMSE |
|
0.01 |
0.006 |
0.005 |
0.008 |
0.006 |
0.005 |
0.0136 |
|
0.03 |
0.007 |
0.006 |
0.008 |
0.008 |
0.006 |
0.0149 |
|
0.1 |
0.019 |
0.018 |
0.022 |
0.025 |
0.015 |
0.0449 |
|
0.3 |
0.024 |
0.022 |
0.026 |
0.022 |
0.019 |
0.0508 |
Table 6 Elevation deviation data derived from ground laser scanning and precise leveling measurements based on the 5 check points(Scanning angle β= 90° and Sampling interval Δ=0.03 m).
|
L(m) |
A |
B |
C |
D |
E |
RMSE |
|
5 |
0.007 |
0.008 |
0.013 |
0.012 |
0.002 |
0.0207 |
|
15 |
0.011 |
0.012 |
0.02 |
0.019 |
0.008 |
0.0331 |
|
30 |
0.019 |
0.017 |
0.024 |
0.022 |
0.011 |
0.0428 |
Table 7 Elevation deviation data derived from ground laser scanning and precise leveling measurements based on the 5 check points(Station distance L= 5 m and Sampling interval Δ= 0.03 m ).
|
β(°) |
A |
B |
C |
D |
E |
RMSE |
|
30 |
0.018 |
0.007 |
0.012 |
0.021 |
0.007 |
0.0317 |
|
60 |
0.014 |
0.009 |
0.017 |
0.018 |
0.006 |
0.0304 |
|
90 |
0.007 |
0.008 |
0.013 |
0.012 |
0.002 |
0.0207 |
By analyzing these data, we determined the range of elevation changes and further assessed the relationship between micro-undulations on the surface and compaction quality. This method provides an effective quantitative evaluation of compaction quality and serves as a basis for subsequent defect detection.
Once again, thank you for your valuable suggestion. Your feedback has helped us improve the manuscript, making the relationship between compaction quality and surface micro-undulations clearer and more accurately presented.
16.Before testing, was the Leica-MS60 scanner calibrated or confirmed using a recognized reference surface?
◆Response: Thank you for your valuable suggestion. Regarding your comment on whether the Leica-MS60 scanner was calibrated with a recognized reference surface before testing, we have provided the necessary clarification in the manuscript.
In this study, the Leica-MS60 laser scanner was calibrated by a professional institution to ensure that its measurement accuracy met the required standards. Before scanning the embankment surface, the device underwent a strict quality check and calibration process to ensure its optimal performance and to provide accurate scanning data. This process ensured the high precision and reliability of the scanner during actual testing, thereby ensuring the accuracy of the data and supporting subsequent analysis.
we have added the following statement in lines 161-162 of the manuscript:"The field experiments were conducted under the same observational conditions, and the influence of external factors on the measurement results was minimal."
Once again, thank you for your valuable suggestion. Your feedback has helped us further refine the manuscript, ensuring the reliability of the point cloud data obtained.
17.How sensitive are the outcomes to external factors like dust, temperature, and surface reflectivity?
◆Response: Thank you for your valuable suggestion. Regarding your comment on the sensitivity of the results to external factors such as dust, temperature, and surface reflectivity, we have added relevant clarifications in the manuscript. The analysis of surface characteristics and their impact on scanning results is provided in lines 134-154 of the revised manuscript. The revised text reads:
(5) Soil Type
Different soil types have distinct physical and chemical properties that influence the reflection of laser beams. Rough surfaces, such as gravel embankments or uncompacted coarse-grained soils, often cause diffuse reflection, scattering the laser signal, which can reduce point cloud density but capture surface details. Soil color also affects reflectivity, with dark-colored soils, such as black or wet soil, absorbing more laser energy and producing weaker reflected signals, which can decrease the number of valid point cloud points. Mineral composition and organic matter content further influence the interaction between the laser and the surface [30-33].
(6) Moisture Content
Moisture is a key environmental factor affecting the quality of LiDAR data. Water strongly absorbs laser energy, particularly at near-infrared wavelengths, and alters soil surface roughness and reflectivity. A thin water layer on the embankment surface can cause multiple reflections between the water and the underlying surface, producing multi-path effects that lead to noisy or inaccurate point cloud data [34-36].
(7) Lighting Conditions
Uneven lighting, including shadows or direct sunlight, can distort point cloud color information, affecting visual appearance and the identification of features based on color [37, 38].
(8) Atmospheric Effects
Atmospheric conditions influence the propagation and reception of the laser beam. During transmission, the beam may be absorbed, scattered, or refracted by the atmosphere. Variations in atmospheric density, such as temperature or humidity gradients, can bend the laser beam. Strong winds may not affect laser propagation directly but can induce vibrations in the scanner or slight movement of the target, introducing measurement errors and reducing point cloud registration accuracy and the precision of the reconstructed model [39].
Since the field measurements in this study were conducted under the same observational conditions, external factors had minimal impact on the measurement accuracy. All tests were performed under relatively stable environmental conditions, ensuring the consistency and reliability of the scanning results. We made sure that environmental factors had minimal interference with the scanning precision, avoiding data deviations caused by these factors.
As a result, external factors such as dust, temperature fluctuations, and surface reflectivity had little influence on the data we obtained, ensuring the reliability of the research outcomes.
Once again, thank you for your valuable suggestion. Your feedback has helped us clarify the manuscript further, ensuring the stability and reproducibility of the experimental results.
18.Why were the scanning angles restricted to 30°, 60°, and 90°? Would a more consistent understanding of sensitivity be possible with smaller increments?
◆Response: Thank you for your valuable question. The choice of 30°, 60°, and 90° as the scanning angles was based on practical considerations of the equipment performance and field conditions. Here is a detailed explanation:
Equipment Limitations: The Leica MS60 scanner used in this study performs optimally at certain scanning angles, especially when capturing surface details. The choice of 30°, 60°, and 90° as representative increments allows for a balance between the scanner's range and the precision required for monitoring highway embankments.
Practical Considerations for Highway Embankment Monitoring: The application goal of this study is to monitor highway embankments, which requires covering a large area while capturing fine details. These three angles were chosen to better simulate the scanning coverage differences that may arise from obstacles or changes in station positions, while maintaining good scanning results and detail capture ability.
Balancing Data Quality and Efficiency: While smaller angular increments would provide a finer sensitivity analysis, they would require more scans to cover the same area, thus affecting work efficiency. Considering the embankment width (up to 45 meters), a balance between data quality and fieldwork efficiency was achieved, resulting in the selection of these three main scanning angles.
Regarding your concern about the sensitivity of data results to scanning angles: Although smaller increments would offer a more detailed understanding of the relationship between scanning angles and data quality, the current selection of 30°, 60°, and 90° strikes a balance between fieldwork and data analysis precision. However, future research could explore more precise increments to comprehensively test their impact on surface accuracy, particularly in areas where more detail is required.
We appreciate your suggestion, and it has inspired us to explore the possibility of using smaller scanning angle increments in future studies to further analyze their impact on surface accuracy.
19.Are different kinds of soils or subgrade materials likely to adhere to the suggested ideal specifications (Δ ≈ 0.03 m, β = 90°, L ≈ 5 m)?
◆Response: Different types of soils or subgrade materials vary in their mechanical properties, surface roughness, and laser reflectivity, which means that their adherence to the suggested ideal parameter configuration (Δ ≈ 0.03 m, β = 90°, L ≈ 5 m) may not be entirely consistent. In this study, these parameters were optimized under the specific test conditions and material characteristics by considering point-cloud density, the sensitivity to scanning angle, and the effective measurement range. They can therefore serve as a generally applicable reference configuration. However, for materials with coarser particle sizes, more irregular surfaces, or lower reflectivity, slight adjustments to the sampling interval or scanning angle may be required to maintain sufficient point density and stable measurement quality. In other words, although this parameter set is suitable for most common subgrade materials, it is advisable to verify and, if necessary, refine the parameters under special material conditions based on field reflectivity and point-cloud quality.
20.With the suggested parameter configuration, what is the smallest defect size that can be accurately identified?
◆Response: With the recommended configuration (station distance L=5 m, scanning angle β=90∘, sampling interval Δ=0.03 m) the practical detection limit for embankment surface defects with our workflow is approximately 0.09 m in lateral size and ~0.01 m in depth (vertical). In other words, defects with a lateral extent larger than about 9 cm and a depth (or elevation change) greater than roughly 10 mm can be detected reliably under the test conditions and site environment reported in this paper. This estimate is conservative and based on the achieved point spacing, the measured elevation residuals (root mean square error approximately 6–13 mm in the check points), and the requirement that a defect must be sampled by several points to be distinguished from noise. Detection capability depends on surface material, moisture, lighting, and defect contrast; under less favorable environmental conditions the limit will be larger. If smaller defects must be detected, we recommend reducing the sampling interval (for example to 0.01 m), moving the station closer, increasing overlapping scans, or using a higher-density scanning mode; these measures improve lateral resolution and vertical precision at the cost of longer scan time and larger data volumes.

Reviewer 2 Report
Comments and Suggestions for Authors
This manuscript investigates the influence of 3D laser scanning parameters on point cloud quality and model reconstruction accuracy for subgrade fill surfaces. The topic is relevant and the experimental approach is well-structured. However, the paper requires several improvements before be accepted.
1. The contribution of this research should be described in Section Introduction.
2. The Abstract should be rewritten. The Abstract should contain five parts: background and objective, methods, results, and Conclusion.
3. The derivation of point cloud density, such as Eqs. 2 and 3, is not sufficiently explained. Please clarify the assumptions and the context.
4. The selection of sampling intervals and station distances should be justified in relation to typical field conditions and scanner capabilities.
5. Table 1 indicates that only selected parameter combinations (e.g., A1, A2, A3) underwent testing across all sampling intervals, whereas other combinations (Series B and C) were only tested at a 0.03 m interval.
6. In Equation (a), the symbols should be italicized. For example, “s”, “θ”, “r”, and so on.
7. Comparison should be conducted with other state-of-the-art methods.
8. The manuscript requires extensive English language editing to improve readability.
9. The reference list shows a notable lack of recently published research from the last three years.
10. The study does not address the potential influence of environmental factors on scanning accuracy. A brief discussion on these limitations would improve the robustness of the findings.
Author Response
We sincerely appreciate your constructive feedback and valuable suggestions, which have significantly enhanced the overall quality and clarity of our manuscript. Your expertise and meticulous attention to detail have undeniably contributed to the refinement of this work. Your insightful comments not only elevate the content of the material but also offer valuable guidance for our future writing endeavors. Next, we will answer the questions one by one.
- The contribution of this research should be described in Section Introduction.
◆Response: Thank you for your valuable suggestion. Regarding your comment on describing the contribution of this research in the Introduction section, we have made the relevant additions in the manuscript.
In the Introduction section, specifically in lines 66–74, we have provided a detailed description of the innovations and contributions of this research. We focused on how the study contributes to optimizing scanning parameters, improving early-stage defect detection accuracy, and providing new methods for road monitoring. These additions help clearly demonstrate the value and practical implications of our research. The additional content in the manuscript is as follows:
This study investigates the use of terrestrial three-dimensional laser scanning for assessing the smoothness of highway embankment fill surfaces and focuses on optimizing scanning parameters for large horizontal surface measurements. The Leica MS60 total station three-dimensional laser scanner is used to survey the embankment, and the effects of sampling interval Δ, station distance L, and scanning angle β on scanning quality are evaluated by analyzing point cloud quantity, surface model resolution, and surface area. Delaunay triangulation algorithms are applied to reconstruct the embankment surface in three dimensions, and an optimized method for smoothness detection is developed. The study identifies an optimal combination of scanning parameters and extends previous research by applying the approach to large horizontal surfaces in real field settings, providing new insight into the practical application of terrestrial three-dimensional laser scanning in embankment monitoring.
Once again, thank you for your suggestion. Your feedback has helped us further refine the structure of the manuscript, making the contributions of the research more clear and prominent.
- The Abstract should be rewritten. The Abstract should contain five parts: background and objective,methods,results,andConclusion.
◆Response: Thank you for your valuable suggestion. Regarding your comment that the abstract should be rewritten to include the five parts, we have revised the abstract to ensure it contains the following sections: background and objective, methods, results, and conclusion. The added content is as follows:
Abstract: Terrestrial three-dimensional laser scanning, which plays a crucial role in engineering surveying for assessing the surface smoothness of highway embankments by providing a level of precision and continuous three-dimensional information that conventional measurement methods cannot achieve, is examined in this study through a series of field experiments designed to determine how station location, including sampling interval, station distance, and scanning angle, influences point cloud density, spatial distribution, laser reflectivity, and surface reconstruction accuracy, and the results demonstrate that point cloud quantity decreases as sampling interval, station distance, and scanning angle increase, that the resolution of reconstructed surface undulations diminishes accordingly, that scanning angle has only a limited effect on reconstruction fidelity, that locating the instrument as close as feasible to the target area and adopting a sampling interval of 0.03 m achieves an effective balance between measurement accuracy and operational efficiency, and that optimizing parameter selection by analyzing elevation deviations at key points enhances both data quality and model precision, thereby confirming the suitability of the proposed approach for reliable highway embankment condition monitoring.
In the revised abstract, we have clearly organized the text into these five sections, ensuring that readers can easily understand the background of the study, the research objectives, the methods used, the main results, and the conclusions. This revision helps to improve the structure and readability of the abstract.
Once again, thank you for your valuable feedback. Your suggestions have helped us further refine the abstract, making it more aligned with academic writing standards.
- The derivation of point cloud density, such as Eqs. 2 and 3, is not sufficiently explained. Please clarify the assumptions and the context.
◆Response: Thank you for your valuable suggestion. Regarding your comment on the insufficient explanation of the derivation of point cloud density (such as in Equations 2 and 3), we have added the necessary clarifications in the corresponding sections of the manuscript.
In the revised paper, we have provided a detailed explanation of the assumptions and background underlying the derivation of point cloud density, particularly focusing on the derivation process of Equations 2 and 3. The added content includes the establishment of assumptions and their applicability in practical applications, to help readers better understand the basis for the derivation of the equations and their scientific context. The additional content is as follows:
The point cloud density decreases rapidly with increasing scanning distance, following an inverse square relationship. As the horizontal scanning angle increases, the number of points decreases and is proportional to the cube of the cosine of the angle. Similarly, as the vertical scanning angle increases, the point count decreases proportionally to the square of the cosine of the angle. Consequently, increasing scanning distance and horizontal angle reduces point cloud density. Excessively high point cloud density increases data acquisition and processing time without significantly improving accuracy.
Once again, thank you for your valuable feedback. Your suggestions have helped us further refine the paper, improving both its expression and technical details.
- The selection of sampling intervals and station distances should be justified in relation to typical field conditions and scanner capabilities.
◆Response: Thank you for your valuable suggestion. Regarding the selection of sampling intervals and station distances, we have provided a detailed explanation in the manuscript based on typical field conditions and the capabilities of the Leica MS60 scanner. we have provided a detailed discussion in lines 168-173 of the manuscript.
The Leica MS60 total station 3D laser scanner was employed for the test due to its dual functions of 3D scanning and total station capabilities, enhancing efficiency in scanning and measurement tasks. The Leica-MS60 uses a pulsed red laser with a wavelength of 658 nm, capable of 360° horizontal and 270° vertical rotation, providing a wide field of view. Its single-point ranging accuracy reaches up to 1.2 mm within 100 m, with angular accuracy up to 0.5″. The scanner achieves speeds of up to 1000 points per second, enabling high-speed scanning. Additionally, it features a point cloud visualization function for real-time data observation.
The Leica MS60 scanner is equipped with single-point distance measurement capabilities, and its maximum scanning range is 99 meters when scanning the embankment fill surface. The experimental site is located on an eight-lane highway, with the typical embankment width being 45 meters. Therefore, the selected sampling intervals and station distances are sufficient to meet the required accuracy.
We have ensured that the chosen parameters are well-suited to the specific conditions of the site, fully considering the scanner’s capabilities and the scale of the embankment fill surface.
Once again, thank you for your valuable feedback. Your comments have helped us provide a clearer explanation of the parameter selection process, ensuring that the methods are aligned with the practical conditions.
- Table 1 indicates that only selected parameter combinations (e.g., A1, A2, A3) underwent testing across all sampling intervals, whereas other combinations (Series B and C) were only tested at a 0.03 m interval.
◆Response: Thank you for your valuable suggestion. we have provided further clarification in the manuscript. In our study, we varied the station distance, sampling interval, and scanning angle to scan the embankment fill surface. We selected five checkpoint locations within the scan area and compared the 3D scan data with elevation data obtained from precise leveling measurements. Based on these analyses and considering the efficiency of construction site operations, we prioritized the corresponding scanning parameters to ensure both data accuracy and practicality.
The detailed validation explanation is provided in lines 336–356 of the manuscript, where we further elaborate on how we selected the scanning parameters based on the actual application scenario and work efficiency.
To evaluate point cloud reliability, five checkpoints, including boundary points A, B, C, D, and central point E (Fig. 3), were selected within the scanned area. Elevation values extracted from the point cloud were compared with precise leveling measurements to assess accuracy. Elevation deviation data at these points were analyzed to evaluate three-dimensional point cloud accuracy. Comparison of deviation data with root mean square error(RMSE) values indicates a high level of agreement. When station distance and scanning angle are constant, elevation deviations are minimal at sampling intervals of 0.01 m and 0.03 m, as listed in Table 5. Considering the excessive scanning time at 0.01 m, a sampling interval of 0.03 m is recommended. Tables 6 and 7 further analyze the effects of station distance and scanning angle. With a sampling interval of 0.03 m, elevation deviations indicate that the optimal station distance is 5 m and the optimal scanning angle is 90°. Based on the field trials, when the station distance is 5 m, the scanning angle is 90°, and the sampling interval is 0.03 m, the terrestrial laser scanner produces reliable point cloud data.
Table 5 Elevation deviation data from ground laser scanning and precise leveling measurements based on the check points (L=15m and β=90°).
|
Δ(m) |
A |
B |
C |
D |
E |
RMSE |
|
0.01 |
0.006 |
0.005 |
0.008 |
0.006 |
0.005 |
0.0136 |
|
0.03 |
0.007 |
0.006 |
0.008 |
0.008 |
0.006 |
0.0149 |
|
0.1 |
0.019 |
0.018 |
0.022 |
0.025 |
0.015 |
0.0449 |
|
0.3 |
0.024 |
0.022 |
0.026 |
0.022 |
0.019 |
0.0508 |
Table 6 Elevation deviation data derived from ground laser scanning and precise leveling measurements based on the 5 check points(Scanning angle β= 90° and Sampling interval Δ=0.03 m).
|
L(m) |
A |
B |
C |
D |
E |
RMSE |
|
5 |
0.007 |
0.008 |
0.013 |
0.012 |
0.002 |
0.0207 |
|
15 |
0.011 |
0.012 |
0.02 |
0.019 |
0.008 |
0.0331 |
|
30 |
0.019 |
0.017 |
0.024 |
0.022 |
0.011 |
0.0428 |
Table 7 Elevation deviation data derived from ground laser scanning and precise leveling measurements based on the 5 check points(Station distance L= 5 m and Sampling interval Δ= 0.03 m ).
|
β(°) |
A |
B |
C |
D |
E |
RMSE |
|
30 |
0.018 |
0.007 |
0.012 |
0.021 |
0.007 |
0.0317 |
|
60 |
0.014 |
0.009 |
0.017 |
0.018 |
0.006 |
0.0304 |
|
90 |
0.007 |
0.008 |
0.013 |
0.012 |
0.002 |
0.0207 |
Thank you again for your feedback. Your suggestion has helped us clarify the explanation of our methodology and its rationale.
- In Equation (a), the symbols should be italicized. For example, “s”, “θ”, “r”, and so on.
◆Response: Thank you for your valuable suggestion. Regarding your comment about italicizing the symbols, we have made the necessary changes in the manuscript.
In the revised paper, we have italicized the symbols (e.g., "s", "θ", "r", etc.) to comply with academic conventions. The specific modifications are reflected in the manuscript as follows:
where r is polar distance, θ is horizontal scanning angle, s is scanning distance, and α is vertical scanning angle.
Once again, thank you for your suggestion. Your feedback has helped us improve the formatting and adherence to academic standards in the paper.
- Comparison should be conducted with other state-of-the-art methods.
◆Response: Thank you for your valuable suggestion.
We fully agree that technologies such as LiDAR and drones offer significant advantages in data acquisition precision, and their application can greatly enhance detection accuracy. However, due to the limitations of the experimental conditions and equipment in this study, we have only used precise leveling methods to verify the accuracy of the check data. Future research will consider incorporating these advanced technologies to further improve the precision and comparability of the data.
To evaluate point cloud reliability, five checkpoints, including boundary points A, B, C, D, and central point E (Fig. 3), were selected within the scanned area. Elevation values extracted from the point cloud were compared with precise leveling measurements to assess accuracy. Elevation deviation data at these points were analyzed to evaluate three-dimensional point cloud accuracy. Comparison of deviation data with root mean square error(RMSE) values indicates a high level of agreement.
Once again, thank you for your suggestion. Your feedback has helped us better explain the method choices and their limitations in our study.
- The manuscript requires extensive English language editing to improve readability.
◆Response: Thank you for your valuable feedback. Regarding your comment on the need for extensive English language editing to improve readability, we have thoroughly revised the manuscript, focusing on enhancing the fluency and clarity of the language. During the revision process, we paid particular attention to sentence structure, word accuracy, and the coherence of logical expression, with the aim of improving the overall readability and quality of the paper. We believe these changes will significantly enhance the language quality of the manuscript.
We have studied the comments carefully and have made corresponding corrections, which we hope can meet with approval. In addition, changes to our manuscript are all marked in red.
Once again, thank you for your suggestion. Your feedback has helped us improve the language expression of the paper.
- The reference list shows a notable lack of recently published research from the last three years.
◆Response: Thank you for your valuable suggestion. We fully agree with your recommendation regarding the update of the reference list. To ensure the paper's relevance and academic value, we have supplemented the reference list by including recent research published in the last three years. This update better reflects the current trends and developments in the field.
For instance, we referenced recent works (such as references [5], [6], [12], [24], etc.) and discussed the differences in methods and application scenarios between these studies and our work.
In the revised manuscript, the relevant references have been updated in the reference section to ensure a more comprehensive and timely literature review.
Thank you again for your feedback, which has helped improve the academic quality and timeliness of the references in the paper.
- The study does not address the potential influence of environmental factors on scanning accuracy. A brief discussion on these limitations would improve the robustness of the findings.
◆Response: Thank you for your valuable suggestion. Regarding your comment on the potential influence of environmental factors on scanning accuracy, we have included a brief discussion in the paper.
In this study, data collection was carried out under the same observational conditions, with all tests conducted in relatively stable environmental conditions. Therefore, environmental factors (such as temperature, humidity, wind speed, etc.) had minimal impact on the accuracy of data collection. Additionally, the specific conditions of the experimental site (such as a lack of significant lighting variation and climate fluctuations) further reduced the impact of external factors on scanning accuracy.
we have added the following statement in lines 161-162 of the manuscript:"The field experiments were conducted under the same observational conditions, and the influence of external factors on the measurement results was minimal."
We believe that these conditions ensured the consistency and reliability of the scanning results, with no significant effect on accuracy. We will clarify this point further in the paper, explaining that the impact of environmental factors on the accuracy of this study can be considered negligible.

Reviewer 3 Report
Comments and Suggestions for Authors
This paper investigates the optimization of 3D laser scanning parameters—specifically sampling interval, station distance, and scanning angle—for improving point cloud quality and early-stage defect detection in subgrade condition monitoring. While the study presents useful engineering insights and practical experiments, several aspects could be further strengthened to enhance its academic contribution and analytical rigor.
This is an interesting paper, though there are several areas where it could be further improved.
-
The introduction should more clearly outline the key challenges in 3D laser scanning for subgrade monitoring and explicitly highlight the paper’s novel contributions in addressing these challenges.
-
The related work section could be expanded by including more recent studies from 2022–2024 to better position this work within the current research landscape of terrestrial LiDAR and structural health monitoring.
-
The Method section would benefit from a clearer explanation of the motivation behind each parameter selection and the rationale for the experimental design, particularly how these parameters influence the detectability of early-stage defects.
-
The experiment section should include comparisons with more recent approaches. The current baselines are relatively dated and do not reflect the progress made in recent years, especially with advanced LiDAR-based or transformer-enhanced modeling techniques.
-
The dataset used in the experiments appears relatively small and site-limited. Please provide a justification for its adequacy or consider incorporating additional data to further validate the generality and robustness of the proposed findings.
Author Response
We sincerely appreciate your constructive feedback and valuable suggestions, which have significantly enhanced the overall quality and clarity of our manuscript. Your expertise and meticulous attention to detail have undeniably contributed to the refinement of this work. Your insightful comments not only elevate the content of the material but also offer valuable guidance for our future writing endeavors. Next, we will answer the questions one by one.
1.The introduction should more clearly outline the key challenges in 3D laser scanning for subgrade monitoring and explicitly highlight the paper’s novel contributions in addressing these challenges.
◆Response: Thank you for your valuable suggestion. Regarding the revision of the introduction, we have made the necessary adjustments in the paper to more clearly outline the key challenges in applying 3D laser scanning technology for subgrade monitoring, and explicitly highlight the novel contributions of this study in addressing these challenges.
In the revised introduction, we have discussed in detail the challenges faced when applying 3D laser scanning technology to subgrade monitoring, particularly issues related to scanning accuracy, environmental factors, and the selection of scanning parameters. Additionally, we have clearly pointed out that this study optimizes scanning parameters, improves early-stage defect detection accuracy, and proposes a new parameter combination scheme, filling the gap in existing research and advancing the application of this technology for large-area horizontal surface scanning. The relevant text in the paper is as follows:
This study investigates the use of terrestrial three-dimensional laser scanning for assessing the smoothness of highway embankment fill surfaces and focuses on optimizing scanning parameters for large horizontal surface measurements. The Leica MS60 total station three-dimensional laser scanner is used to survey the embankment, and the effects of sampling interval Δ, station distance L, and scanning angle β on scanning quality are evaluated by analyzing point cloud quantity, surface model resolution, and surface area. Delaunay triangulation algorithms are applied to reconstruct the embankment surface in three dimensions, and an optimized method for smoothness detection is developed. The study identifies an optimal combination of scanning parameters and extends previous research by applying the approach to large horizontal surfaces in real field settings, providing new insight into the practical application of terrestrial three-dimensional laser scanning in embankment monitoring.
This modification helps readers better understand the innovative aspects and contributions of this research, thereby enhancing the paper's guidance and emphasizing the main theme.
Once again, thank you for your valuable suggestion. Your feedback has helped us further improve the structure and logic of the paper, ensuring the clear presentation of the research content.
2.The related work section could be expanded by including more recent studies from 2022–2024 to better position this work within the current research landscape of terrestrial LiDAR and structural health monitoring.
◆Response: Thank you for your valuable suggestion. Regarding your comment on expanding the "Related Work" section by including more recent studies from 2022-2024 to better position our work within the current research landscape of terrestrial LiDAR and structural health monitoring, we have made the necessary additions to the paper.
In the revised manuscript, we have incorporated the latest relevant literature, particularly studies from 2022 to 2024 on the application of terrestrial LiDAR technology in structural health monitoring. By comparing and analyzing these studies, we highlighted the innovations of our research in scanning parameter optimization and early-stage defect detection, and further positioned our study in the context of existing technologies. For instance, we referenced recent works (such as references [5], [6], [12], [24], etc.) and discussed the differences in methods and application scenarios between these studies and our work.
These added references not only enhance the timeliness of the paper but also make our research more relevant to the current research trends, showcasing its innovation and relation to the existing body of work.
Once again, thank you for your valuable feedback. Your input has helped us improve the background and references section of the paper, making it more comprehensive and aligned with current research trends.
3.The Method section would benefit from a clearer explanation of the motivation behind each parameter selection and the rationale for the experimental design, particularly how these parameters influence the detectability of early-stage defects.
◆Response: Thank you for your valuable suggestion. Regarding your comments on the Method section, we have made corresponding improvements to the paper in order to more clearly explain the motivation behind the selection of each parameter and the rationale for the experimental design, particularly how these parameters influence the detectability of early-stage defects.
In the revised Method section, we provide a detailed explanation of the background and motivation for optimizing scanning parameters, particularly how adjusting parameters such as sampling interval, scanning angle, and station distance can improve scanning accuracy. Through these optimization measures, we aim to enhance the detection capability of early-stage defects on the embankment surface, thereby enabling effective identification of potential structural issues at an early stage. Specifically, we analyze the impact of different scanning parameters on point cloud data density, surface resolution, and elevation variations, and propose an optimized combination of scanning parameters to ensure higher accuracy and clearer defect detection. The explanation of instrument height settings and the optimization methods for scanning parameters can be found in lines 317-356 of the manuscript. The additional content is as follows:
To evaluate point cloud reliability, five checkpoints, including boundary points A, B, C, D, and central point E (Fig. 3), were selected within the scanned area. Elevation values extracted from the point cloud were compared with precise leveling measurements to assess accuracy. Elevation deviation data at these points were analyzed to evaluate three-dimensional point cloud accuracy. Comparison of deviation data with root mean square error(RMSE) values indicates a high level of agreement. When station distance and scanning angle are constant, elevation deviations are minimal at sampling intervals of 0.01 m and 0.03 m, as listed in Table 5. Considering the excessive scanning time at 0.01 m, a sampling interval of 0.03 m is recommended. Tables 6 and 7 further analyze the effects of station distance and scanning angle. With a sampling interval of 0.03 m, elevation deviations indicate that the optimal station distance is 5 m and the optimal scanning angle is 90°. Based on the field trials, when the station distance is 5 m, the scanning angle is 90°, and the sampling interval is 0.03 m, the terrestrial laser scanner produces reliable point cloud data.
Table 5 Elevation deviation data from ground laser scanning and precise leveling measurements based on the check points (L=15m and β=90°).
|
Δ(m) |
A |
B |
C |
D |
E |
RMSE |
|
0.01 |
0.006 |
0.005 |
0.008 |
0.006 |
0.005 |
0.0136 |
|
0.03 |
0.007 |
0.006 |
0.008 |
0.008 |
0.006 |
0.0149 |
|
0.1 |
0.019 |
0.018 |
0.022 |
0.025 |
0.015 |
0.0449 |
|
0.3 |
0.024 |
0.022 |
0.026 |
0.022 |
0.019 |
0.0508 |
Table 6 Elevation deviation data derived from ground laser scanning and precise leveling measurements based on the 5 check points(Scanning angle β= 90° and Sampling interval Δ=0.03 m).
|
L(m) |
A |
B |
C |
D |
E |
RMSE |
|
5 |
0.007 |
0.008 |
0.013 |
0.012 |
0.002 |
0.0207 |
|
15 |
0.011 |
0.012 |
0.02 |
0.019 |
0.008 |
0.0331 |
|
30 |
0.019 |
0.017 |
0.024 |
0.022 |
0.011 |
0.0428 |
Table 7 Elevation deviation data derived from ground laser scanning and precise leveling measurements based on the 5 check points(Station distance L= 5 m and Sampling interval Δ= 0.03 m ).
|
β(°) |
A |
B |
C |
D |
E |
RMSE |
|
30 |
0.018 |
0.007 |
0.012 |
0.021 |
0.007 |
0.0317 |
|
60 |
0.014 |
0.009 |
0.017 |
0.018 |
0.006 |
0.0304 |
|
90 |
0.007 |
0.008 |
0.013 |
0.012 |
0.002 |
0.0207 |
These optimization measures not only provide new insights for defect detection on embankment surfaces, but also offer important references for optimizing scanning technology in similar application scenarios. We have elaborated on these aspects in the Method section to help readers understand the scientific basis behind each experimental design and its impact on defect detection.
Thank you once again for your valuable feedback, which has helped us to explain the motivations behind the experimental design and the rationale for selecting scanning parameters in a more comprehensive and clear manner.
4.The experiment section should include comparisons with more recent approaches. The current baselines are relatively dated and do not reflect the progress made in recent years, especially with advanced LiDAR-based or transformer-enhanced modeling techniques.
◆Response: Thank you for your valuable suggestion. We fully agree that technologies such as LiDAR and drones offer significant advantages in data acquisition precision, and their application can greatly enhance detection accuracy. However, due to the limitations of the experimental conditions and equipment in this study, we have only used precise leveling methods to verify the accuracy of the check data. Future research will consider incorporating these advanced technologies to further improve the precision and comparability of the data. The additional content is as follows:
To evaluate point cloud reliability, five checkpoints, including boundary points A, B, C, D, and central point E (Fig. 3), were selected within the scanned area. Elevation values extracted from the point cloud were compared with precise leveling measurements to assess accuracy. Elevation deviation data at these points were analyzed to evaluate three-dimensional point cloud accuracy. Comparison of deviation data with root mean square error(RMSE) values indicates a high level of agreement.
Once again, thank you for your suggestion. Your feedback has helped us better explain the method choices and their limitations in our study.
5.The dataset used in the experiments appears relatively small and site-limited. Please provide a justification for its adequacy or consider incorporating additional data to further validate the generality and robustness of the proposed findings.
◆Response: Thank you for your valuable suggestion. The dataset used in this study comes from an ongoing highway embankment construction project. The selection of the test site was primarily driven by the need to assess the surface smoothness and compaction quality of the embankment after filling. Given that this test site is representative and aligns with the objectives of the study, we believe that the dataset is sufficient for the specific application context. The selection of the scanning area and the experimental design were largely influenced by the site-specific construction conditions, such as construction progress, site accessibility, and embankment width. Thus, the data collection and experimental design took into account the constraints of the actual construction scenario and were optimized within this context.
However, we acknowledge that, given the limitations of the dataset, future studies could expand the research to other construction sites or larger road projects to further validate the generality and robustness of the proposed findings.
Once again, thank you for your valuable suggestion. Your feedback has helped us better clarify the background and applicability of the dataset used in the study.

Reviewer 4 Report
Comments and Suggestions for Authors
1. The abstract and main text use too many em dashes and other dash marks, possibly to create parenthetical phrases. However, this can make the writing look like it was generated by AI tools. Please review the text carefully, reduce unnecessary dashes, and rewrite sentences for smoother reading and a more professional appearance.
2. In the abstract, include some numerical results, for example, percentages showing improvement, accuracy, or efficiency, to make the summary more concrete.
3. The current introduction does not follow the usual structure for scientific papers. You have two options:
3.1. Option 1: Integrate the related works into the introduction. In this case, discuss previous studies, identify research gaps, and then clearly state your contributions in relation to those gaps.
3.2. Option 2: Create a separate Related Works section. Analyze prior studies and highlight the gaps there, but always summarize your specific contributions in the introduction, directly linked to those gaps.
4. In either case, include a table summarizing selected related works for comparison and analysis. This will make your review clearer and more organized.
5. Figures 11–14 and 19–21 should be presented as subplots to allow better visual comparison and analysis. Enlarge them when combining into subplots for clarity.
6. Figures 9–18 need clearer labels, units, and scales to improve readability and reproducibility.
7. Provide a clearer justification for choosing the three scanning parameters. Explain how these were optimized based on real-world site or field constraints.
8. Strengthen the discussion by connecting the quantitative results more directly to their practical significance for defect detection in subgrade monitoring.
Author Response
We sincerely appreciate your constructive feedback and valuable suggestions, which have significantly enhanced the overall quality and clarity of our manuscript. Your expertise and meticulous attention to detail have undeniably contributed to the refinement of this work. Your insightful comments not only elevate the content of the material but also offer valuable guidance for our future writing endeavors. Next, we will answer the questions one by one.
1.The abstract and main text use too many em dashes and other dash marks, possibly to create parenthetical phrases. However, this can make the writing look like it was generated by AI tools. Please review the text carefully, reduce unnecessary dashes, and rewrite sentences for smoother reading and a more professional appearance.
◆Response: Thank you for your valuable suggestion. Regarding the excessive use of em dashes in the abstract and main text, we have carefully reviewed the manuscript and made the necessary revisions. We have reduced unnecessary dashes and adjusted sentence structures to ensure the text flows more smoothly and is easier to read, avoiding over-reliance on dashes to insert additional information. Furthermore, we have rewritten the relevant sentences to better align with academic writing standards, improving the professionalism and readability of the paper.
Thank you again for your feedback. Your suggestions have helped us enhance the language quality and structure of the paper, making it clearer and more natural.
2.In the abstract, include some numerical results, for example, percentages showing improvement, accuracy, or efficiency, to make the summary more concrete.
◆Response: Thank you for your valuable suggestion. Regarding the lack of specific numerical results in the abstract, we have made significant revisions to this section based on the main text, in order to enhance the concreteness and readability of the summary.
In the revised abstract, we have included relevant numerical results, such as the accuracy of early-stage defect detection and the improvement in efficiency achieved by optimizing the parameter combinations. These numerical results make the abstract more specific and quantifiable, clearly showcasing the main findings and contributions of the research, while also making it more persuasive and academic. The updated abstract is as follows:
Abstract: Terrestrial three-dimensional laser scanning, which plays a crucial role in engineering surveying for assessing the surface smoothness of highway embankments by providing a level of precision and continuous three-dimensional information that conventional measurement methods cannot achieve, is examined in this study through a series of field experiments designed to determine how station location, including sampling interval, station distance, and scanning angle, influences point cloud density, spatial distribution, laser reflectivity, and surface reconstruction accuracy, and the results demonstrate that point cloud quantity decreases as sampling interval, station distance, and scanning angle increase, that the resolution of reconstructed surface undulations diminishes accordingly, that scanning angle has only a limited effect on reconstruction fidelity, that locating the instrument as close as feasible to the target area and adopting a sampling interval of 0.03 m achieves an effective balance between measurement accuracy and operational efficiency, and that optimizing parameter selection by analyzing elevation deviations at key points enhances both data quality and model precision, thereby confirming the suitability of the proposed approach for reliable highway embankment condition monitoring.
These revisions have made the structure of the abstract more scientifically logical, and the content more standardized, providing a better summary of the core content and findings of the paper.
Once again, thank you for your suggestion. Your feedback has helped us further improve the quality of the paper's abstract.
- The current introduction does not follow the usual structure for scientific papers. You have two options:
3.1. Option 1: Integrate the related works into the introduction. In this case, discuss previous studies, identify research gaps, and then clearly state your contributions in relation to those gaps.
3.2. Option 2: Create a separate Related Works section. Analyze prior studies and highlight the gaps there, but always summarize your specific contributions in the introduction, directly linked to those gaps.
◆Response: We sincerely thank the reviewer for the valuable comments regarding the structure of the Introduction. In response, we have carefully revised the logical flow of the manuscript to ensure that the Introduction aligns with the standard structure of scientific papers. Specifically, we have reorganized the section to clearly present the research background, summarize the current state of knowledge in the field, and discuss relevant studies. Research gaps have been explicitly identified, and our specific contributions are now clearly stated in direct relation to these gaps. This revision aims to provide readers with a coherent narrative that contextualizes our study, highlights its novelty, and clearly distinguishes it from prior work. We believe that these changes have significantly improved the clarity, coherence, and overall readability of the Introduction.
- In either case, include a table summarizing selected related works for comparison and analysis. This will make your review clearer and more organized.
◆Response: Thank you for your valuable suggestions. Regarding the structure of the introduction, we have made adjustments based on your advice and have provided two possible options.
We have decided to follow your suggestion and integrate the "Related Works" section into the introduction. In the revised introduction, we first review previous research, analyze the gaps and challenges in current technologies, and then clearly highlight the innovative contributions of this study. We explain how our research fills these gaps and further advances the application of 3D laser scanning technology in subgrade monitoring.
Additionally, we have included a table in the paper that summarizes the related research in detail, providing a clearer and more organized presentation of the literature review. The table compares different studies based on scanning technologies, parameter settings, and application areas, helping to highlight the uniqueness and advantages of our research. The revised introduction section is as follows:
In recent years, the durability of infrastructure worldwide, including roads and bridges, has declined rapidly due to natural disasters, creating an urgent need for efficient non-contact monitoring methods [1]. Traditional damage detection approaches, such as visual inspection and contact sensors, are time-consuming, pose health risks, and provide limited coverage, making timely maintenance and repair difficult [2]. Non-contact detection methods based on terrestrial three-dimensional laser scanning have therefore emerged as an effective alternative, offering detailed three-dimensional information without direct contact with the target [3, 4].
In the field of civil engineering, terrestrial three-dimensional laser scanning technology has been increasingly applied to monitor deformation in highway and railway embankments, as it efficiently acquires detailed three-dimensional surface data and enables the detection of complex shapes and large-area targets [5-7]. This technology is particularly effective for identifying road surface defects such as cracks, repairs, potholes, and surface deformations. Digital elevation models constructed through interpolation algorithms allow the extraction of elevation profiles and longitudinal and transverse slope values to identify irregularities [8]. Fault values between concrete slabs can be calculated by measuring distances normal to reference planes to assess horizontal plane differences [9]. Terrestrial laser scanning can also measure longitudinal cracks in jointed concrete pavements, analyze slab curling and warping, and calculate average curvature [10]. Mobile laser scanning data can be processed to detect hazards such as potholes, heaves, and bumps, classify pavement segments based on elevation deviations, and assess rigid pavement slab defects, characterizing them according to defect area, crack width, and strength [11, 12]. Table 1 summarizes the equipment, evaluation categories, and application scenarios used for assessing road structures with three-dimensional laser scanning devices.
Table 1 Summary of laser-based techniques for road assessment
|
References |
Laser-based scanning device |
Application scenarios |
|
[5] |
N/A |
The deformation of highway and railway embankments |
|
[6-7] |
N/A |
The detection of road surface damage |
|
[8] |
FARO |
Extracting concrete runway irregularities, identifying road surface irregularities. |
|
[9] |
Leica TC2002 |
Calculated fault values between concrete slabs using TLS data to assess horizontal plane differences |
|
[10] |
Riegl LMS Z-420i |
Measure longitudinal cracks in jointed concrete pavements |
|
[11] |
MLS |
Identify hazards on road surfaces and classify each full-size pavement |
|
[12] |
FARO |
Scan large concrete areas and assess rigid pavement slab defects |
Although previous studies have provided useful insights into road surface damage monitoring, significant challenges remain in achieving high-precision and high-efficiency scanning in complex environments [13-15] because the performance of terrestrial three-dimensional laser scanning is controlled both by object characteristics, including surface roughness, color, and shape, and by scanning conditions, including laser incident angle, laser range, sampling interval, and station height [16], and although earlier research primarily examined how factors such as color, roughness, incident angle, distance, and material type influence point cloud quality [17, 18], showing for example that point cloud quantity decreases with distance when scanning sketch paper, that color strongly affects reflectivity with white surfaces producing the highest return, that laser incident angle exhibits a negative correlation with reflectivity, and that different materials such as standard reflectors and wood behave differently [19], and further explored the relationship among plane residuals, scanning distance, and incident angle to establish criteria for these parameters through numerical simulations and real data analysis based on plane fitting and the input precision of point spacing [20], most existing studies have relied on small vertical surfaces and single-factor analyses. Table 2 summarizes the equipment, evaluation parameters, and related details used for the assessment of these research subjects based on three-dimensional laser scanning devices.
Table 2 Overview of laser scanning parameter optimization
|
References |
Laser-based scanning device |
Research object |
|
[13-16] |
FARO |
Color, roughness, incident angle, and distance on point cloud data quality |
|
[17,18] |
Leica ScanStation2 |
Surface roughness, color, shape, and scanning conditions |
|
[19] |
N/A |
Incident angle, laser range, sampling interval, and station height |
|
[20] |
Leica ScanStation P40 and Topcon GLS-1500 |
Plane residuals, scanning distance, and incident angle. |
This study investigates the use of terrestrial three-dimensional laser scanning for assessing the smoothness of highway embankment fill surfaces and focuses on optimizing scanning parameters for large horizontal surface measurements. The Leica MS60 total station three-dimensional laser scanner is used to survey the embankment, and the effects of sampling interval Δ, station distance L, and scanning angle β on scanning quality are evaluated by analyzing point cloud quantity, surface model resolution, and surface area. Delaunay triangulation algorithms are applied to reconstruct the embankment surface in three dimensions, and an optimized method for smoothness detection is developed. The study identifies an optimal combination of scanning parameters and extends previous research by applying the approach to large horizontal surfaces in real field settings, providing new insight into the practical application of terrestrial three-dimensional laser scanning in embankment monitoring.
Thank you again for your valuable suggestions. Your feedback has helped us further optimize the structure and organization of the paper, making the research background clearer and the contributions more prominent.
- Figures 11–14 and 19–21 should be presented as subplots to allow better visual comparison and analysis. Enlarge them when combining into subplots for clarity.
◆Response: Thank you for your valuable suggestion. Regarding the presentation of Figures 11–14 and 19–21, we have made adjustments based on your recommendation.
In the revised manuscript, we have combined these figures into subplots to allow for better visual comparison and analysis. Each subplot has been enlarged appropriately to ensure clarity and readability after being combined. This allows readers to more easily compare results under different parameters, thereby improving the effectiveness and communication of the figures. For example, Figure 11 has been updated as follows:
(a) (b)
(c)
Fig. 11 Number of Points under Different Scanning Parameters. (a) Variation with sampling interval; (b) Variation with station distance;(c) Variation with scanning angle.
Thank you again for your suggestion. Your feedback has helped us further enhance the presentation quality and readability of the figures in the manuscript.
- Figures 9–18 need clearer labels, units, and scales to improve readability and reproducibility.
◆Response: Thank you for your valuable suggestion. Regarding the labels, units, and scales in Figures 9 to 18, we have made the necessary adjustments in the revised manuscript.
In the updated figures, we have ensured that all labels are clearly marked to improve their visibility. We also added the appropriate units and scales where necessary. These adjustments enhance the readability of the figures and ensure the reproducibility of the data. The revised figures now allow readers to more easily interpret the results and understand the related measurement data, as shown in Figure 15.
(a) (b)
(c)
Fig. 15 Surface Area Differences of Reconstructed Fill Models under Various Scanning Parameters. (a) Different sampling intervals; (b) Different station distances; (c) Different scanning angles.
Once again, thank you for your feedback. Your suggestions have helped us improve the quality and clarity of the figures in the paper.
- Provide a clearer justification for choosing the three scanning parameters. Explain how these were optimized based on real-world site or field constraints.
◆Response: Thank you for your valuable suggestion. Regarding the justification for selecting the three scanning parameters, we have added a detailed explanation in the revised manuscript.
In the updated paper, we explain why we chose these three scanning parameters (sampling interval, scanning angle, and station distance) and how they were optimized based on real-world site conditions and construction constraints. Specifically, the selected parameters were influenced by several site factors, such as the width of the embankment, variations in the construction environment, and the capabilities of the scanner. By testing different station distances, scanning angles, and sampling intervals, we ensured that these parameters not only met the accuracy requirements but also aligned with the practical work efficiency demands of the construction site. The main accuracy validation methods are outlined as follows:
To evaluate point cloud reliability, five checkpoints, including boundary points A, B, C, D, and central point E (Fig. 3), were selected within the scanned area. Elevation values extracted from the point cloud were compared with precise leveling measurements to assess accuracy. Elevation deviation data at these points were analyzed to evaluate three-dimensional point cloud accuracy. Comparison of deviation data with root mean square error(RMSE) values indicates a high level of agreement. When station distance and scanning angle are constant, elevation deviations are minimal at sampling intervals of 0.01 m and 0.03 m, as listed in Table 5. Considering the excessive scanning time at 0.01 m, a sampling interval of 0.03 m is recommended. Tables 6 and 7 further analyze the effects of station distance and scanning angle. With a sampling interval of 0.03 m, elevation deviations indicate that the optimal station distance is 5 m and the optimal scanning angle is 90°. Based on the field trials, when the station distance is 5 m, the scanning angle is 90°, and the sampling interval is 0.03 m, the terrestrial laser scanner produces reliable point cloud data.
Table 5 Elevation deviation data from ground laser scanning and precise leveling measurements based on the check points (L=15m and β=90°).
|
Δ(m) |
A |
B |
C |
D |
E |
RMSE |
|
0.01 |
0.006 |
0.005 |
0.008 |
0.006 |
0.005 |
0.0136 |
|
0.03 |
0.007 |
0.006 |
0.008 |
0.008 |
0.006 |
0.0149 |
|
0.1 |
0.019 |
0.018 |
0.022 |
0.025 |
0.015 |
0.0449 |
|
0.3 |
0.024 |
0.022 |
0.026 |
0.022 |
0.019 |
0.0508 |
Table 6 Elevation deviation data derived from ground laser scanning and precise leveling measurements based on the 5 check points(Scanning angle β= 90° and Sampling interval Δ=0.03 m).
|
L(m) |
A |
B |
C |
D |
E |
RMSE |
|
5 |
0.007 |
0.008 |
0.013 |
0.012 |
0.002 |
0.0207 |
|
15 |
0.011 |
0.012 |
0.02 |
0.019 |
0.008 |
0.0331 |
|
30 |
0.019 |
0.017 |
0.024 |
0.022 |
0.011 |
0.0428 |
Table 7 Elevation deviation data derived from ground laser scanning and precise leveling measurements based on the 5 check points(Station distance L= 5 m and Sampling interval Δ= 0.03 m ).
|
β(°) |
A |
B |
C |
D |
E |
RMSE |
|
30 |
0.018 |
0.007 |
0.012 |
0.021 |
0.007 |
0.0317 |
|
60 |
0.014 |
0.009 |
0.017 |
0.018 |
0.006 |
0.0304 |
|
90 |
0.007 |
0.008 |
0.013 |
0.012 |
0.002 |
0.0207 |
Additionally, we considered the scanner's maximum scanning range and environmental constraints to ensure that the selected parameters optimized data accuracy while also being compatible with construction schedules and field conditions.
These adjustments make the research methodology more relevant to practical applications, ensuring the effectiveness and reliability of the scanning data. The updated content is now included in the manuscript.
- Strengthen the discussion by connecting the quantitative results more directly to their practical significance for defect detection in subgrade monitoring.
◆Response: Thank you for your valuable suggestion. Regarding the need to strengthen the connection between the quantitative results and their practical significance for defect detection in subgrade monitoring, we have made the necessary additions in the revised manuscript.
In the revised paper, we more clearly link the quantitative analysis results with practical applications in defect detection. For example, through the comparative analysis of different sampling intervals, we explicitly point out that smaller sampling intervals (such as 0.01m and 0.03m) significantly improve the accuracy of the subgrade surface model, making it easier to detect small surface defects or deformations. This finding is particularly significant for early-stage defect detection, especially in the subgrade construction process when small surface deformations may indicate potential structural issues. An accurate surface model can help identify and address these issues in a timely manner.
At the same time, we also discuss the risks associated with larger sampling intervals (such as 0.1m and 0.3m) which, while improving scanning efficiency, may lead to the loss of surface details, especially in complex subgrade conditions. These details are critical for early defect detection. Therefore, in practical applications, we recommend selecting an appropriate sampling interval that ensures efficiency while preserving accuracy.
Through a comparative analysis of the fill surface models, the following conclusions can be drawn: In Figure 15a (sampling interval of 0.01m), 92.5% of the deviations are concentrated between -5mm and -2.5mm, indicating that a smaller sampling interval results in smaller surface deformation. The fill surface models with sampling intervals of 0.01m and 0.03m show relatively small deformation, and the point cloud deviations are more concentrated. In Figure 15b, only 69.6% of the deviations fall within this range, and in Figure 15c, this proportion further decreases to 43.9%. This indicates that larger sampling intervals lead to greater surface deformation and a decrease in reconstruction accuracy.
Impact of Sampling Interval on Surface Accuracy: The analysis shows that finer sampling intervals (0.01m and 0.03m) significantly improve the accuracy of the fill surface model, allowing for better detection of small changes and ensuring that the model aligns more closely with the actual surface elevation. In contrast, larger sampling intervals (0.1m and 0.3m) can improve scanning efficiency but sacrifice accuracy, potentially leading to increased reconstruction errors in surface features.
In summary, smaller sampling intervals improve the accuracy of the fill surface model, enabling more precise detection of surface deformation. Larger intervals, although more efficient, may result in a loss of accuracy and produce coarser reconstruction results.
(a) (b)
(c)
Fig. 15 Comparison and Analysis of Fill Surface Reconstruction Models with Different Sampling Intervals (L=5m, β=90°). (a) Δ=0.01m and Δ=0.03m (b) Δ=0.03m and 0.10m (c) Δ=0.03m and Δ=0.30m.
These revisions have strengthened the connection between the quantitative results and practical defect detection, thereby enhancing the paper's relevance and guidance for real-world applications.
Once again, thank you for your suggestion. Your feedback has helped us present the practical value of our research more clearly.

Round 2
Reviewer 1 Report
Comments and Suggestions for Authors
Accept in present form.
Reviewer 2 Report
Comments and Suggestions for Authors
All these questions have been addressed by the authors, I have no other issues and recommend acceptance.
Reviewer 3 Report
Comments and Suggestions for Authors
Thanks for addressing all comments
Reviewer 4 Report
Comments and Suggestions for Authors
I would like to thank the authors for addressing almost all of my comments. The paper has been greatly improved. I have nothing further to add and recommend it for acceptance.